# Cost-effectiveness of incorporating Ebola prediction score tools and rapid diagnostic tests into a screening algorithm: A decision analytic model

Antoine Oloma Tshomba[1,2]*, Daniel Mukadi-Bamuleka[2,3], Anja De Weggheleire[4], Olivier M. Tshiani[2,3], Charles T. Kayembe[5], Placide Mbala-Kingebeni[2,3], Jean-Jacques Muyembe-Tamfum[2,3], Steve Ahuka-Mundeke[2,3], Faustin M. Chenge[1,6], Bart Karl M. Jacobs[7], Dieudonné N. Mumba[2,4], Désiré D. Tshala-Katumbay[2,8,9], Sabue Mulangu[2,3]

1 Department of Public Health, University of Kisangani, Kisangani, Democratic Republic of Congo (DRC), 2 National Institute for Biomedical Research, Kinshasa, DRC, 3 Department of Medical Biology, University of Kinshasa, Kinshasa, DRC, 4 Department of Tropical Medicine, University of Kinshasa, Kinshasa, DRC, 5 Department of Internal Medicine, University of Kisangani, Kisangani, DRC, 6 School of Public Health, University of Lubumbashi, Lubumbashi, RDC, 7 Department of Clinical Sciences, Institute of Tropical Medicine, Antwerp, Belgium, 8 Department of Neurology and School of Public Health, Oregon Health & Science University, Portland, Oregon, United States of America, 9 Department of Neurology, University of Kinshasa, Kinshasa, DRC

* antotshomba@yahoo.fr

**Data Availability Statement:** All relevant data are within the paper.

## Abstract

### Background

No distinctive clinical signs of Ebola virus disease (EVD) have prompted the development of rapid screening tools or called for a new approach to screening suspected Ebola cases. New screening approaches require evidence of clinical benefit and economic efficiency. As of now, no evidence or defined algorithm exists.

### Objective

To evaluate, from a healthcare perspective, the efficiency of incorporating Ebola prediction scores and rapid diagnostic tests into the EVD screening algorithm during an outbreak.

### Methods

We collected data on rapid diagnostic tests (RDTs) and prediction scores' accuracy measurements, e.g., sensitivity and specificity, and the cost of case management and RDT screening in EVD suspect cases. The overall cost of healthcare services (PPE, procedure time, and standard-of-care (SOC) costs) per suspected patient and diagnostic confirmation of EVD were calculated. We also collected the EVD prevalence among suspects from the literature. We created an analytical decision model to assess the efficiency of eight screening strategies: 1) Screening suspect cases with the WHO case definition for Ebola suspects, 2) Screening suspect cases with the ECPS at -3 points of cut-off, 3) Screening suspect cases with the ECPS as a joint test, 4) Screening suspect cases with the ECPS as a conditional

**Funding:** This study has received partial support from NIH (Grant reference NIH FIC/R01EY031894). The funder had no role in the design of the study; in the collection, analyses, or interpretation of data; in the writing of the manuscript, or in the decision to publish the results.

**Competing interests:** The authors declare that they have no competing interests

test, 5) Screening suspect cases with the WHO case definition, then QuickNavi™-Ebola RDT, 6) Screening suspect cases with the ECPS at -3 points of cut-off and QuickNavi™-Ebola RDT, 7) Screening suspect cases with the ECPS as a conditional test and Quick-Navi™-Ebola RDT, and 8) Screening suspect cases with the ECPS as a joint test and QuickNavi™-Ebola RDT. We performed a cost-effectiveness analysis to identify an algorithm that minimizes the cost per patient correctly classified. We performed a one-way and probabilistic sensitivity analysis to test the robustness of our findings.

## Results

Our analysis found dual ECPS as a conditional test with the QuickNavi™-Ebola RDT algorithm to be the most cost-effective screening algorithm for EVD, with an effectiveness of 0.86. The cost-effectiveness ratio was 106.7 USD per patient correctly classified. The following algorithms, the ECPS as a conditional test with an effectiveness of 0.80 and an efficiency of 111.5 USD per patient correctly classified and the ECPS as a joint test with the QuickNavi™-Ebola RDT algorithm with an effectiveness of 0.81 and a cost-effectiveness ratio of 131.5 USD per patient correctly classified. These findings were sensitive to variations in the prevalence of EVD in suspected population and the sensitivity of the Quick-Navi™-Ebola RDT.

## Conclusions

Findings from this study showed that prediction scores and RDT could improve Ebola screening. The use of the ECPS as a conditional test algorithm and the dual ECPS as a conditional test and then the QuickNavi™-Ebola RDT algorithm are the best screening choices because they are more efficient and lower the number of confirmation tests and overall care costs during an EBOV epidemic.

## Introduction

Since the mid-nineties, Ebola virus disease (EVD) outbreaks have emerged in sub-Saharan tropical Africa, where about 130 million people live with filovirus exposure risk [1–5]. The infection is plagued by a high case fatality rate (CFR), which ranges between 50% and 90%.

The control of EVD outbreaks relies on the early and accurate implementation of public health measures such as 1) surveillance and detection of suspect cases; 2) ring vaccination and tracing of contacts; 3) prompt isolation of cases and care management, and 4) infection prevention and control measures (household decontamination and safe and dignified burials). The care management of patients with EVD includes standard-of-care (SOC), nutrition, and specific therapeutics [6]. Recently, two monoclonal antibody-based therapies received FDA approval for treating EVD, strongly changing the prognosis of the infection by reducing the lethality below 35% [7].

During EVD outbreaks, surveillance relies on the World Health Organization (WHO) clinical case definition to admit suspect cases at the point-of-care and transfer them to the care unit while waiting for EVD laboratory confirmation. After EVD confirmation, specific treatment is administered to confirmed cases in addition to the SOC. EVD diagnosis relies on the GeneXpert® Ebola Assay (Cepheid, Sunnyvale, CA, USA) [8], an automated, sensitive, and specific reverse transcriptase polymerase chain reaction (RT-PCR) technology.

However, GeneXpert® is still expensive and requires trained staff, a power supply, minimal infrastructure, and a reliable supply chain to be able to use it at the peripheral levels where most EVD outbreaks occur in poor resource settings [9–11]. While in theory the turnaround time is 2–4 hours, in reality delays of 2–5 days have been noted [12, 13]. In addition, the WHO clinical case definition used in the field to screen for EVD is insufficiently accurate to discriminate EVD cases from non-cases at the point-of-care [14–16].

As most common tropical diseases such as malaria, typhoid fever, meningitis, etc. can display EVD-like symptoms, non-EVD cases may also reach the care units for clinical management and further interventions. In this context, it is more likely that false positives are isolated, raising the workload, and undermining the availability of scarce resources.

For the above reasons (i.e., the cost, technical demand, and long turnaround time for the EVD results, coupled with the poor discriminating performance of the WHO clinical case definition used), the WHO issued a target product profile (TPP) for EBOV tests, including rapid diagnostic tests (RDTs), to shorten the turnaround time and provide high accuracy (desired level: sensitivity >98%, specificity >99%; acceptable level: sensitivity > 95%, specificity > 99%) [11, 17, 18]. Subsequently, researchers have developed many Ebola (EBOV) antigen-based rapid diagnostic tests (RDTs) and RT-PCR assays [19, 20]. Some of these screening tools are under evaluation or have already been evaluated and showed better performance [20–22].

Simultaneously, some researchers have developed clinical prediction scores as rapid diagnostic tools to assess the disease probability in suspected EVD cases [23–25]. A screening score tool developed is the extended clinical prediction score (ECPS), which includes clinical and epidemiological predictors. This prediction score showed quite good diagnostic accuracy (AUROC of 0.88, 95% CI: 0.86–0.89) and a cross-validated area under the ROC curve (AUCCV) of 0.87 in cross-validation evaluation [26]. The authors proposed three scenarios for the implementation of the ECPS: 1) the score by choosing an operational cut-off for action; 2) the ECPS as a joint test; and 3) the score as a conditional test to target additional diagnostic testing.

For instance, the WHO recommendation for the use of rapid, sensitive, safe, and simple Ebola diagnostic tests to optimize EVD screening underlying innovative approaches for screening Ebola suspect cases [17, 27]. Thus, developing novel screening methods requires evidence of clinical benefit and economic efficiency to identify cost-effective strategies to use in the screening of EVD suspect cases, which can help policymakers determine whether RDT and diagnostic prediction scores could efficiently replace the WHO case definition for Ebola suspect cases at the point-of-care. Thus far, there is no evidence of the cost-effectiveness and efficiency of rapid diagnostic tools.

This study assesses, from a healthcare perspective, the cost-effectiveness of combining Ebola rapid diagnostic tests (RDTs) and prediction tools in the screening of EVD cases based on a decision analytic model.

## Methods

Decision analysis is a quantitative approach to evaluate the consequences of alternative strategies and to guide the choice of the most effective or cost-effective course of action under uncertainty [28]. Decision analysis requires a decision tree that identifies every possible decision and the consequence of each decision and then assigns a probability and a payoff to each consequence [29].

We considered the healthcare system perspective for this analysis, and all analyses were performed using TreeAge Pro software version 2021 (TreeAge, Williamstown, Massachusetts, USA).

**Table 1. Description of screening algorithms compared in the model.**

| Screening algorithm | Algorithm description |
|---|---|
| **Algorithm 1** | Screening EVD suspect cases with WHO case definition |
| **Algorithm 2** | Screening EVD suspect cases with the ECPS at -3 points of cut-off |
| **Algorithm 3** | Screening EVD suspect cases using ECPS as a joint test or approach |
| **Algorithm 4** | Screening EVD suspect cases using ECPS as a conditional test or approach |
| **Algorithm 5** | Screening EVD suspect cases by combining/sequencing the WHO case definition for suspect cases first and the QuickNavi™-Ebola RDT |
| **Algorithm 6** | Screening EVD suspect cases by combining/sequencing the ECPS at -3 points of cut-off first and the QuickNavi™-Ebola RDT |
| **Algorithm 7** | Screening EVD suspect cases by combining/sequencing the ECPS as a conditional test or approach first and the QuickNavi™-Ebola RDT. |
| **Algorithm 8** | Screening EVD suspect cases by combining/sequencing the ECPS as a joint test or approach first and the QuickNavi™-Ebola RDT. |

Footnotes: ECPS: Extended clinical prediction score, RDT: rapid diagnostic test.

## Screening algorithms

The surveillance used the WHO case definition for EVD to recruit suspect cases ("algorithm 1"). As per the WHO case definition, a suspect case is any person with sudden fever and hemorrhage, or sudden fever with at least three other general symptoms such as severe headache, muscle and joint pain, and fatigue [30]. We compared the algorithm 1 with 1) ECPS at -3 points of cut-off, 2) ECPS as a join test, 3) ECPS as a conditional test, and 4) along with the QuickNavi™-Ebola RDT (e.g., WHO case definition, ECPS at -3 points of cut-off, ECPS as a joint test, or ECPS as a conditional test followed by QuickNavi™-Ebola RDT).

For algorithms with a combination of two screening tests, one positive diagnostic test (e.g., RDT, clinical prediction score, or WHO case definition) must isolate an EVD suspect case. Table 1 describes the different screening algorithms compared in this analysis.

As described by Tshomba et al. [26], the two screening methods—joint and conditional tests with ECPS—are methods in which suspects with no reported risk of exposure would be assumed to be free of the disease, and the clinical team would act appropriately (e.g., no further action is taken). Using the joint approach, all suspects at low-, intermediate-, and high-risk reported exposure are clinically assessed, and only those with a predicted likelihood of EVD greater than 5% are suggested for isolation. In the conditional test, regardless of their estimated probability of contracting the illness, all suspects with high-risk reported exposure should be isolated. Next, suspects with low and intermediate reported exposure who have an EVD-predicted probability more than 5% should be isolated.

## Decision models and outcomes

Fig 1 depicts a decisional tree comparing the related screening-action strategies. From the decision root node, each resulting branch represents the strategy chosen to screen the EVD suspect cases. S1 Fig draws the complete decision tree model, and S1 File describes and defines each algorithm tested in the model (S1 Fig and S1 File).

Decision branches may provide the following outcomes: 1) "EVD case isolated" (a true EVD case (true positive) isolated and clinically managed in temporary healthcare with the SOC); 2) "Non-EVD case erroneously isolated" (a non-Ebola case (false positive) isolated and cared for in temporary healthcare with the SOC); 3) "EVD case erroneously ruled out" (a true

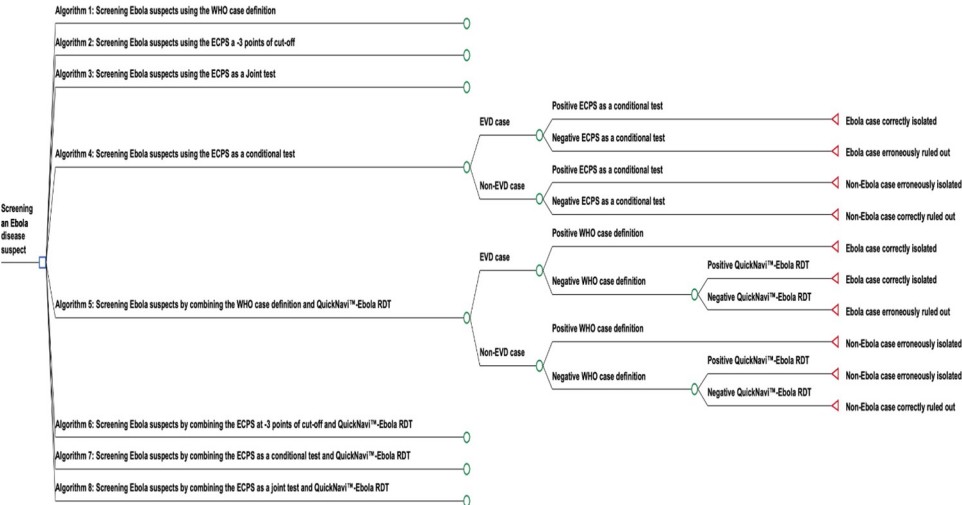

**Fig 1. Decision tree for eight competing algorithms for the screening of Ebola virus disease (EVD).** This is a reduced tree displayed. Not all the branch sequences are displayed in the graph. The non-displayed ones follow the same sequence, e.g., as one test to screen Ebola suspects or combining/sequencing two tests to screen Ebola suspects (thus, the same as the two examples of possible scenarios displayed in the figure). Algorithms 1, 2, and 3 use a single screening test; their visual representations are similar to the branch shown on the decision branch of algorithm 4. Algorithms 6, 7, and 8 use two sequencing screening tests; their visual presentations look like this, shown in this format on algorithm 5's branch. QuickNavi™-Ebola RDT is used after the first screening test in algorithms with two screening tests. EVD = Ebola virus disease; ECPS = extended clinical prediction score; WHO = World health organization.

Ebola case sent back into the community (false negative)); 4) "Non-EVD case correctly ruled out" (a non-Ebola case (true negative) sent back to the community).

## Probability estimates

Table 2 shows the probabilities included in our cost-effectiveness model. The table includes baseline estimates and plausible intervals to be used for the sensitivity analysis. We retrieved probabilities from the literature review or computed them from the DRC's 2018–2020 EVD outbreak's surveillance data. The probabilities included the prevalence of EVD among the suspected population, the sensitivity, and the specificity of the RDT or prediction scores included in the decision model (**Table 2**).

**Table 2. Probability parameters included in the decision tree model.**

| Parameter | Baseline value | Plausible range or 95%CI | Reference source |
|---|---|---|---|
| Prevalence of EVD infection in suspected population (%) | 6.2 | 2.0–10.0 | [26] |
| Prevalence of EVD infection in suspected population (%) | 0.062 | 0.02–0.10 | [26] |
| Sensitivity following the WHO criteria (%) | 81.5 | 74.1–87.2 | [15] |
| Specificity following the WHO criteria (%) | 35.7 | 28.5–43.6 | [15] |
| Sensitivity of ECPS at -3 points of cut-off (%) | 98.0 | 96.5–98.9 | [26] |
| Specificity of ECPS at -3 points of cut-off (%) | 37.0 | 36.0–37.9 | [26] |
| Sensitivity of ECPS as a joint test (%) | 80.0 | 76.8–83.0 | Computed using data published in [26] |
| Specificity of ECPS as a joint test (%) | 81.9 | 81.1–82.7 | Computed using data published in [26] |
| Sensitivity of ECPS as a conditional test (%) | 65.4 | 61.6–69.1 | Computed using data published in [26] |
| Specificity of ECPS as a conditional test (%) | 87.1 | 86.4–87.7 | Computed using data published in [26] |
| Sensitivity of QuickNavi™-Ebola RDT (%) | 87.4 | 63.6–96.8 | [20] |
| Specificity of QuickNavi™-Ebola RDT (%) | 99.6 | 99.3–99.8 | [20] |

**Table 3. Effectiveness payoff assigned to outcomes of the decision tree model.**

| Disease status | Action taken | Disease outcome | Baseline value | Plausible range | Source |
|---|---|---|---|---|---|
| EVD-positive case | Correctly isolated | True-positive | 1 | - | by assumption |
| EVD-negative case | Correctly ruled out | True-negative | 1 | - | by assumption |
| EVD-negative case | Erroneously isolated | False-positive | -0.077 | (-0.124)- (-0.037) | See **S2 File**. |
| EVD-positive case | Not isolated | False-negative | -2.49 | (-2.60)-(-2.38) | [33] |

## Effectiveness

Table 3 presents the effectiveness values included as payoffs for outcomes in the decision tree model. We estimated the effectiveness of each EVD screening algorithm by considering all the steps of EVD suspect case management. Therefore, we quantified the effectiveness of each algorithm in terms of the number of isolated EVD cases (true positives). The result of the Gen-eXpert® Ebola test was considered the reference standard. A positive result confirms an EVD infection and calls for specific care. Therefore, we considered any "case correctly isolated" and "non-case correctly ruled out" as benefits and assigned a "one" mark as the effectiveness value for each true positive or true negative.

For each non-case erroneously isolated, we assigned the score of the isolated iatrogenic case minus the probability of infection given random contact with an Ebola patient, as computed by Gilbert [31]. As a frontline vaccination for healthcare workers would be implemented, we computed this probability of infection using the secondary attack rate (SAR) for direct physical contact of 22.9% (95% CI: 11.6%–34.2%) for those with direct contact but no nursing in the hospital [32].

We negatively assigned this probability reported to the number of non-EVD exposed contacts due to this classification error in the isolation (as a payoff). Negatively because it is the harm caused by isolation, e.g., iatrogenic harm. For this erroneously isolated false positive, we assumed that each isolated false positive and his two family caregivers were non-EVD (e.g., three non-EVD would be exposed in the isolation ward). Thus, a value of -0.077 was assigned to each isolated non-EVD case.

We hypothesized that the community as a whole would be exposed to the Ebola virus infection by these false negatives in the community. Therefore, we assigned a score equal to minus the anticipated number of Ebola cases that this false-negative case—which was ruled out—would produce in the entire susceptible population (e.g., minus the basic reproductive number, the Ro, which accounts for the transmissibility and the typical number of community contacts that this false-negative would harm). In a population that is entirely susceptible, the basic reproduction number is the number of secondary instances that one case would result in.

For each EVD case ruled out, we assigned a value of -2.49, e.g., minus the Ro as estimated by Lewnard [33], as the effectiveness payoff (**Table 3**). S2 File gives the details of iatrogenic probability computation (**S2 File**).

## Costs

We used a micro-costing approach to estimate the operational direct costs. Micro costing is a technique relying on a detailed listing of every resource consumed separately for step-by-step individual action [34]. For laboratory workers, we included the DRC Ministry of Health (MoH) salary scale determined during the 2018–2020 outbreak period. Indeed, we calculated the time spent on the sample collection and analysis (GeneXpert® test). We assumed that all alive suspects gave a blood sample for testing and received extensive supportive care while waiting for the results.

We assigned to each suspect in the care unit a cost of USD 342 including 1) supportive systematic treatment, 2) personal protective equipment, 3) personnel costs as estimated by Bartsch et al. [35], and 4) the cost of surveillance estimated at USD 1.8 [36].

Running a single GeneXpert® test took an average of 107 min. This time excluded: 1) the sample collection process at the care unit; 2) the pre-analytical phase in the laboratory (material preparation, labeling, and notification form); 3) the sample reception and unpacking; 4) the sample inactivation and aliquoting within the glovebox; and 5) results reading and delivery. We fixed the cost of a GeneXpert® test at USD 20, corresponding to the pricing given to subsidized partners [37]. We assigned USD 10 to the QuickNavi™-Ebola RDT [38].

We assumed negligible costs for additional supplies (cryotubes, pipet tips, and other supplies), the cost of Cepheid GeneXpert® platform depreciation over time, and capital costs (costs incurred in the same year).

## Analysis

**Efficiency analysis.** The efficiency of each screening strategy was assessed on the basis of the cost-effectiveness ratio in terms of USD per EVD case isolated. The cost-effectiveness ratio for a given algorithm was calculated using the following formula:

$$\text{Cost} - \text{effectiveness ratio} = \frac{\text{Cost of a given Ebola screening algorithm}}{\text{Effectiveness of that Ebola screening algrithm}} \quad (1)$$

Our primary outcomes were 1) the expected costs per suspected case, 2) the number of confirmed EVD cases isolated, and 3) the cost-effectiveness of the proposed screening algorithms. S2 File describes in detail the technical approach used to compute the total cost of screening suspects, the number of EVD isolated for each screening algorithm, and each probability used in the formula.

We computed the incremental cost-effectiveness ratio (ICER) of isolating one additional EVD case by comparing each alternative algorithm to the best screening algorithm after ranking their effectiveness. The ICER was the incremental cost divided by incremental effectiveness, weighted by the EVD prevalence among the suspects. The resulting cost-effectiveness ratio for each algorithm represents the magnitude of additional health gained (e.g., EVD isolated here) per additional unit of resources spent.

The ICER was calculated using the formula as follows:

$$\text{ICER} = \frac{\text{Cost of a given algorithm} - \text{Cost of the algorithm comparator}}{\text{Effectiveness of a given algrithm} - \text{Cost of the algorithm comparator}} \quad (2)$$

Where the numerator, in the case of Ebola disease, represents the incremental cost, which is the total expense incurred due to an additional health effect, e.g., an isolated EVD case. It is calculated by looking at the additional expenses made throughout the screening process, such as supplies used, for one extra health effect. The denominator represents the incremental effectiveness, which is the increase in the effectiveness of the Ebola screening throughout the screening process.

**Sensitivity analysis.** We performed one-way sensitivity analyses, i.e., deterministic analyses, and probabilistic sensitivity analysis. The exact values of each parameter used in the model are uncertain. We performed a series of one-way sensitivity analyses to evaluate the effect of changes in parameter values over their plausible range on the efficiency ranking of algorithms, e.g., to test the robustness of our ranking conclusion.

The parameters included in the model of algorithms and considered for this sensitivity analysis were: 1) the prior Ebola virus disease probability (e.g., the disease prevalence in the

suspected population); 2) the sensitivities and specificities of RDT and scores; 3) the cost of standard-of-care; 4) the cost of QuickNavi™-Ebola RDT; and 5) the cost of the GeneXpert® test. Additionally, as there are currently no marketed RDTs or therapies for Ebola (they are still in negotiations), we performed a 2-way sensitivity analysis exploring the effects of changing the price of the QuickNavi-Ebola RDT and the price of SOC on algorithm ranking. Three levels of the annual per capita 2021-DRC gross domestic product (2021-DRC GDP), as a willingness-to-pay, were used in this analysis (at one, two, and three times the 2021-DRC GDP).

To evaluate the overall cost-effectiveness sensitivity of the model, we performed a probabilistic sensitivity analysis (PSA) using Monte Carlo simulation. This latter quantifies the degree of confidence in the cost-effectiveness outputs based on uncertainty in the model inputs [39]. We plotted the cost-effectiveness acceptability curve (CEAC) to summarize the impact of parameter uncertainty on the cost-effectiveness outcome, the incremental cost-effectiveness ratio. The CEAC plots, on the horizontal axis, a range of cost-effectiveness thresholds against the probability that the screening algorithm will be cost-effective at that threshold on the vertical axis.

To simulate, we replaced the parameters' point estimates by defining probability distributions for selected decision model parameters. We assumed a beta distribution for all probabilities and a gamma distribution for all nonnegative numeric parameters' values. We set the willingness-to-pay (WTP) threshold at USD 50,000. Lastly, as suggested by the World Health Organization Choosing Interventions that are Cost-Effective (WHO-CHOICE) group, we used the country-specific WTP threshold to identify the cost-effective algorithm [40].

For the DRC, we used the annual per capita 2021-DRC gross domestic product, which was USD 584.1 [41]. The best decision is to choose the algorithm that has the highest ICER and falls just at or below the WTP threshold [40].

### Ethic statements

This study was part of the Ebola outbreak response and disease surveillance in the North-Kivu Ebola outbreak in the Democratic Republic of the Congo and did not constitute human research. This economic evaluation study used published results from the literature to build the decision model. Thus, it did not require ethical approval.

### Results

The model output related to the efficiency of the application of each screening algorithm on a suspect case of EVD is presented in Table 4, which reports the 1) cost and effectiveness of the

**Table 4. Cost, incremental cost, effectiveness, incremental effectiveness, cost-effectiveness ratio, and incremental cost-effectiveness ratio of Ebola screening-action algorithms based on baseline value.**

| Screening algorithm | Cost (USD) | Incremental cost (USD) | Effectiveness (patients correctly classified) | Incremental effectiveness (patients correctly classified) | Efficiency (USD per patient correctly classified) | Incremental cost-effectiveness (USD per additional patient correctly classified) | Algorithm Ranking |
|---|---|---|---|---|---|---|---|
| Algorithm 4 | 88.6 | | 0.80 | | 111.5 | | 1 |
| Algorithm 7 | 91.4 | 2.8 | 0.86 | 0.06 | 106.7 | 44.6 | 2 |
| Algorithm 8 | 106.3 | 14.9 | 0.81 | -0.05 | 131.5 | -308.9* | 3 |
| Algorithm 3 | 118.9 | 27.5 | 0.77 | -0.08 | 153.6 | -331.7* | 4 |
| Algorithm 2 | 250.2 | 158.9 | 0.36 | -0.50 | 696.6 | -319.3* | 5 |
| Algorithm 6 | 252.1 | 160.7 | 0.36 | -0.50 | 697.3 | -324.5* | 6 |
| Algorithm 5 | 259.0 | 167.6 | 0.34 | -0.51 | 752.9 | -326.8* | 7 |
| Algorithm 1 | 274.9 | 183.5 | 0.31 | -0.55 | 885.5 | -335.8* | 8 |

**\*:** absolutely dominated

person screened for the complete screening-action process, 2) incremental cost, 3) incremental effectiveness, 4) incremental cost-effectiveness ratio, and 5) efficiency of each algorithm compared with the most effective screening algorithm studied.

At the baseline point of values, e.g., using the point estimates of each input parameter value, the cost of screening using the WHO case definition was USD 274.9 per patient correctly classified ("Algorithm 1") and USD 250.2 using the ECPS at the -3 cut-off point of the score ("Algorithm 2"). The cost was USD 118.9 for screening with the ECPS as a joint test ("Algorithm 3") and USD 88.9 for screening with the EPCS as a conditional test ("Algorithm 4"). The screening costs per case isolated were increased: 1) from 250.2 to 252.17 USD when using the selective QuickNavi™-Ebola RDT testing after negative ECPS at -3 points of cut-off ("Algorithm 6"); and 2) from 88.9 to 91.4USD after negative ECPS as a conditional test. The screening costs per isolated case were decreased: 1) from 274.9 to 259.0 USD (an incremental USD 15.9 [5.8%] decrease) when using the selective QuickNavi™-Ebola RDT testing after negative WHO case definition ("Algorithm 5") and 2) from 118.9 to 106.3 after negative ECPS as a joint test.

The use of the ECPS conditional test ("Algorithm 4") was the cheapest and decreased the screening costs from USD 274.9 to USD 88.6 per patient correctly classified if compared to the traditional WHO case definition algorithm (an incremental USD 186.3 [67.8%] decrease) (**Table 4**).

We found fewer EVD cases (true positives) when using two algorithms without RDT testing (algorithms 1 and 2) and two dual screening algorithms with RDT testing (algorithms 6 and 5). In contrast, the highest number of patients correctly classified was obtained with dual screening with selective QuickNavi™-Ebola testing after a negative ECPS as a conditional test ("Algorithm 7") or as a joint test ("Algorithm 8") and an ECPS as a joint test or a conditional test ("Algorithm 3" or "Algorithm 4"). However, all six screening algorithms were absolutely dominated by the algorithm using ECPS as a conditional test ("Algorithm 4") and the algorithm sequencing ECPS as a conditional test and the QuickNavi™-Ebola testing ("Algorithm 7").

The traditional algorithm using the WHO case definition for suspects ("Algorithm 1") to screen Ebola suspects had an effectiveness of 0.31. This fraction of effectiveness reflects the number of EVD suspects who were correctly classified after taking into consideration the harm brought on by incorrect classifications. It can be seen as the percentage of patients who were correctly categorized for each patient screened. It costs USD 274.9 per isolated case, with USD 885.5 per patient correctly classified for efficiency. The algorithm using the ECPS as a conditional test and the algorithm with the dual ECPS as a conditional test associated with the QuickNavi™-Ebola RDT were the most cost-effective for EVD suspect screening. Compared to ECPS as a joint test alone ("Algorithm 3"), using the ECPS as a conditional test and the dual ECPS as a conditional test with QuickNavi™-Ebola RDT for screening were associated with an efficiency of USD 111.5 and USD 106.7 per patient correctly classified, respectively.

Fig 2 shows the results of one-way sensitivity analysis. The variations in input parameters, including the prevalence of EVD in suspected population and the sensitivity of the Quick-Navi™-Ebola RDT, changed the analysis ranking or conclusion. For instance, the screening algorithm's efficiency using the ECPS as a conditional test and selective QuickNavi™-Ebola RDT testing after a negative ECPS was about 80.0 and 84.3 USD per patient correctly classified for the prevalence under 4% and about 146.7 and 124.2 USD per patient correctly classified, respectively, for the EVD prevalence at 10%(**Fig 2**, and **S3 Fig**). In addition, the variation in disease prevalence among suspected populations changed the effectiveness and cost of the dominant screening algorithms (**S4 Fig**).

Therefore, the ECPS as a joint or conditional test algorithm had the lowest cost at a prevalence greater than 10% (**S4 Fig**). Our one-way sensitivity analysis also indicates that the

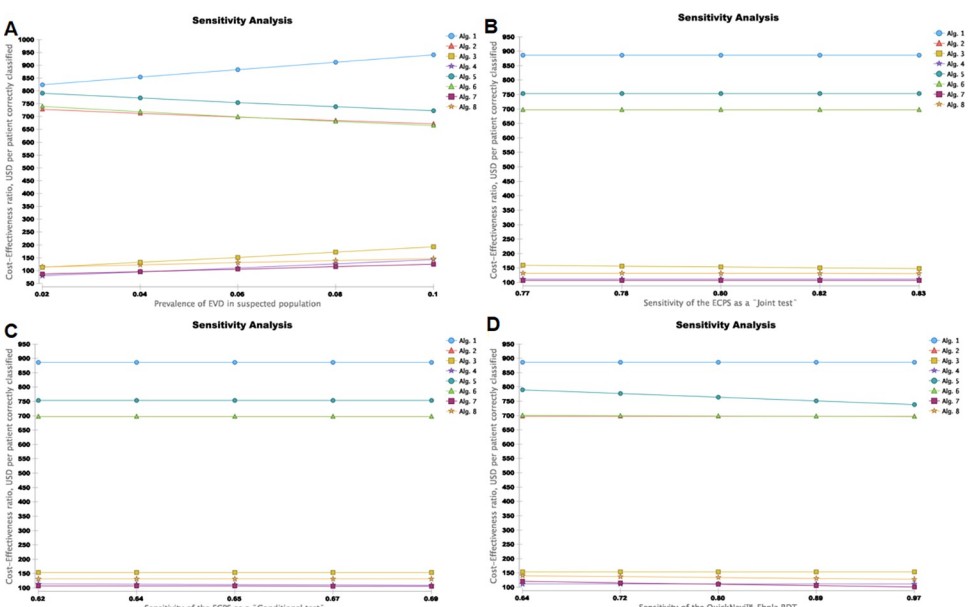

**Fig 2. Variations in cost-effectiveness ratios of eight Ebola screening algorithms as a function of prevalence of Ebola virus disease in suspected population and sensitivities of the ECPS as a joint or conditional test, and the QuickNavi™-Ebola RDT. A** is the effect of variation in the prevalence of the Ebola virus disease on the efficiency of algorithms. **B** is the effect of variation in the sensitivity of the ECPS as a joint test on the efficiency of algorithms. **C** is the effect of variation in the sensitivity of the ECPS as a conditional test on the efficiency of algorithms. **D** is the effect of variation in the sensitivity of the QuickNavi™-Ebola RDT on the Efficiency of algorithms on the Efficiency of algorithms.

prevalence of EVD in the suspected population, the cost of the QuickNavi™— Ebola RDT and the cost of SOC are the most crucial variables that influence the ICER for the dual ECPS as a conditional test with QuickNavi™-Ebola RDT (**Fig 3**).

To address 95% of the total uncertainty of the outcome cost-effectiveness, we should consider the uncertainties for the following parameters: 1) cost of SOC (81%), 2) prevalence of EVD in the suspected population (10%), and 3) cost of QuickNavi™-Ebola RDT (4%).

Fig 4 depicts the two-way sensitivity analysis on the cost of the QuickNavi™-Ebola RDT and the cost of the SOC. S1 Table shows the cost-effectiveness ratios related to the tiered cost of both the QuickNavi™-Ebola RDT and the levels of SOC when using different algorithms. The most cost-effective screening algorithms include the highest number of true positives and the lowest number of false positives by varying the costs of SOC and QuickNavi™-Ebola RDT together. When the cost of the QuickNavi™-Ebola RDT was lower (USD 10), the ranking of the screening algorithms would not change even if the cost of SOC was > USD 150. However, the proportional efficiency estimates would be altered in low cost SOC contexts, e.g., USD 150 per course, if the cost of QuickNavi™-Ebola RDT was > USD 10 (**Fig 4 and S1 Table**).

In probabilistic sensitivity analysis (PSA), the results showed that the dual ECPS as a conditional test with the QuickNavi™-Ebola RDT algorithm displayed the highest probability of being cost-effective among the evaluated algorithms, as shown in Fig 5.

Fig 6 shows the cost-effectiveness acceptability curve and the probability of being cost-effective for each screening algorithm. The screening algorithm with the ECPS as a conditional test ("Algorithm 4") was cost-effective in about 31% of simulations at WTP less than USD 200 and in 0% of simulations at WTP of USD 300 and higher. The probability that dual ECPS as a conditional test with QuickNavi™-Ebola RDT algorithm was the most cost-effective increased from the WTP threshold value of USD 300 and read 100% from the WTP of USD 500 and

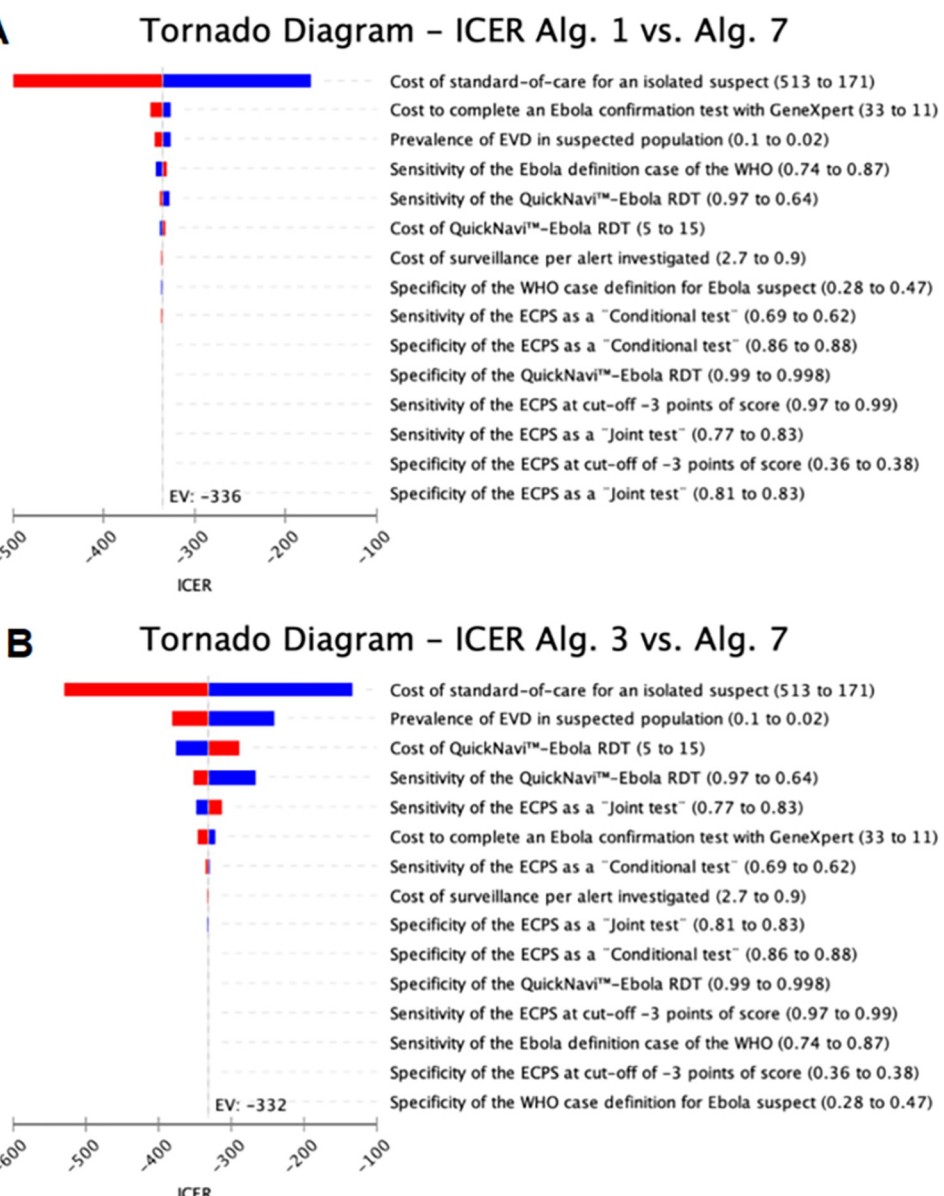

**Fig 3. Tornado diagram presenting One-way sensitivity analysis of ICER comparing the combining ECPS as a conditional test with QuickNavi™— Ebola RDT algorithm (Algorithm 7) to WHO case definition for the suspect algorithm (Algorithm 1) and the ECPS as a joint test algorithm (Algorithm 3).** Vertical line represents incremental effects when using baseline estimates of all parameters. Not all the parameters tested in the sensitivity analysis are visible on the plot. All key variables were included in the sensitivity analysis. Alg. = algorithm; ECPS = extended clinical prediction score; ICER = incremental cost-effectiveness ratio; RDT = rapid diagnostic test; blue: decrease; red: increase.

higher, while this probability was zero or nearly zero for any other algorithms across the WTP threshold spectrum.

At the USD 50 000 WTP threshold, the PSA showed that including single screening testing with the ECPS as a joint test or a conditional test ("Algorithm 3" or "Algorithm 4") and dual screening with selective QuickNavi™-Ebola testing after a negative ECPS as a conditional test ("Algorithm 7") or as a joint test ("Algorithm 8") screening algorithms were cost-effective. Indeed, they were inexpensive and highly effective in 100% of simulations compared to

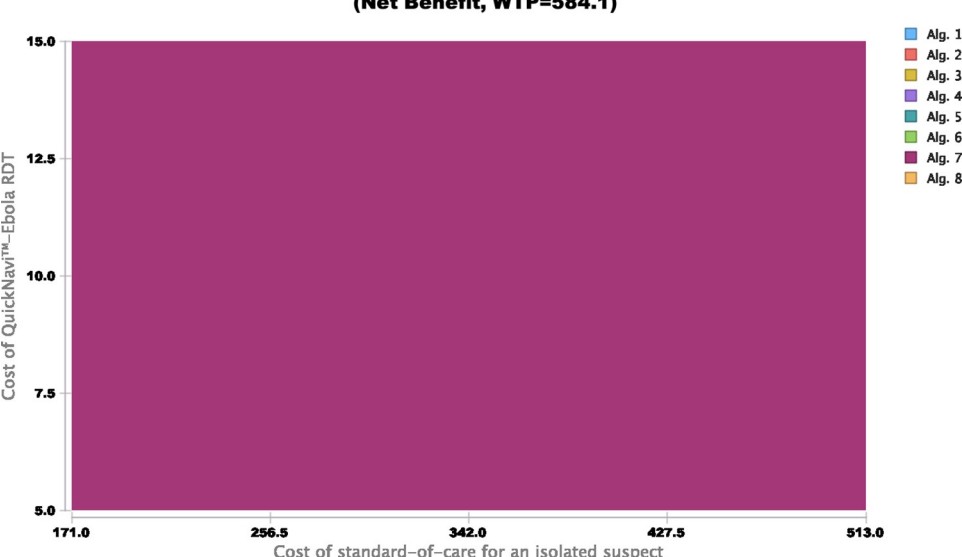

**Fig 4. Two-way sensitivity analysis comparing the net health benefit of EVD screening algorithms by varying both the cost of the QuickNavi™-Ebola RDT and the cost of the standard of care.** The figure shows the two-way sensitivity analysis based on variations in the cost of the QuickNavi™-Ebola RDT and the cost of the SOC at a willingness-to-pay of USD 584.1. For these, a willingness-to-pay of USD 1168.2 and a willingness-to-pay of USD 1752.3 do not appear here, as they display this at a willingness-to-pay of USD 584.1.

traditional screening algorithms with the WHO case definition for the suspects. At this USD 50 000 WPT, the dual screening algorithm with the ECPS at the -3 point cut-off and then the QuickNavi™-Ebola RDT ("Algorithm 6") was cost-effective in 90.3% of simulations; the ECPS at the -3 point cut-off of the score ("Algorithm 2") was cost-effective in 89.7% of simulations;

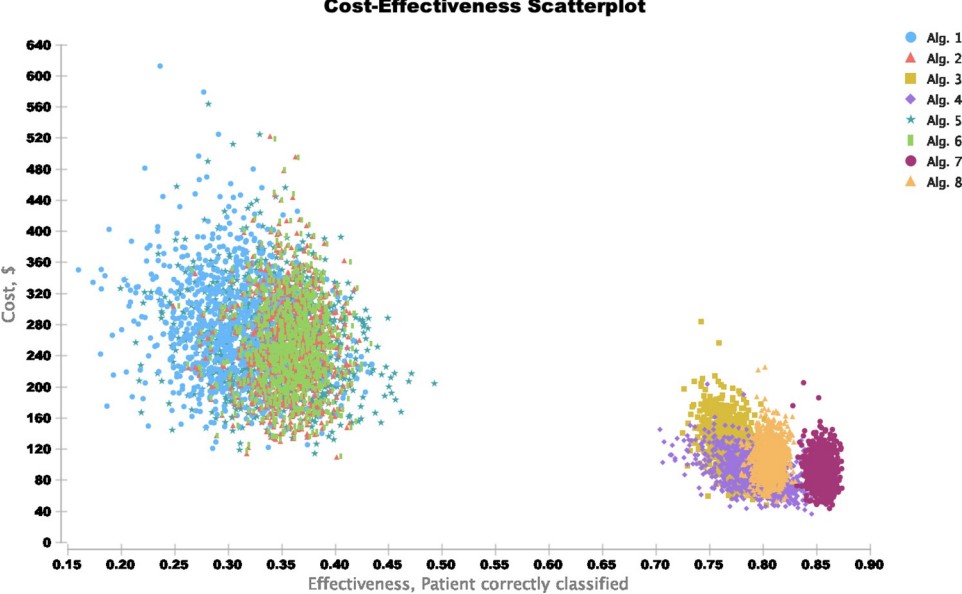

**Fig 5. Cost-effectiveness scatterplot depicting the probabilistic sensitivity analysis (PSA) for 1000 iterations of simulated cost-effectiveness ratio of 8 algorithms for screening Ebola virus disease suspects.**

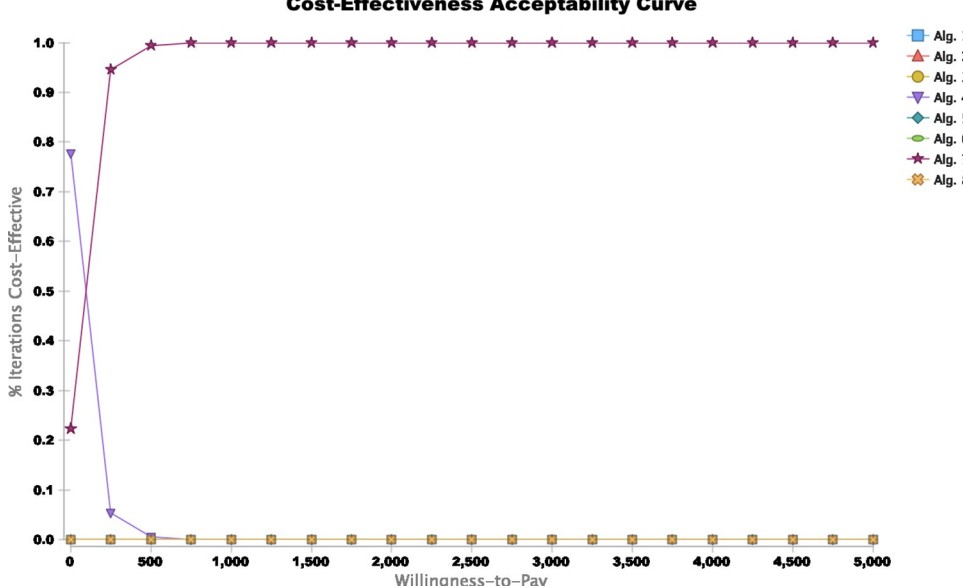

**Fig 6. Cost-effectiveness acceptability curve comparing Algorithm 1 (screening with the WHO case definition) to seven Ebola screening algorithms.** The curves depict the probability of being cost-effective for each screening algorithm. The curves show that the probability that integrating of the dual ECPS as a conditional test with QuickNavi™— Ebola RDT ("Algorithm 7") into the screening algorithm for Ebola suspects compared to any other screening algorithms at varying thresholds WTP. "Algorithm 4" was cost-effective in about 31% of simulations at WTP less than USD 200, and in 0% of simulations at WTP USD 300; "Algorithm 7" was cost-effective in 68.4% of simulations at WTP USD 100, in 97.2% of simulations at WTP USD 350, and in 100% of simulations at WTP of USD 500 and higher. Abbreviations: Alg. = algorithm; EVD = Ebola virus disease; ECPS = extended clinical prediction score; WTP = willingness-to-pay.

and the selective QuickNavi™-Ebola RDT testing after a negative WHO case definition ("Algorithm 5") was in about 100% of simulations (Fig 7). Additionally, the dual screening algorithm with ECPS as a conditional test and QuickNavi™-Ebola RDT was 100% cost-effective in a simulation pattern compared to any other screening algorithm.

Regarding the "WHO case definition" algorithm, at the WTP threshold of GDP per capita (USD 584.1) per additional EVD isolated, we found it cost-effective in 100% simulations while using ECPS as a joint test algorithm, ECPS as a conditional test algorithm, dual ECPS as a joint test and QuickNavi™-Ebola RDT algorithm, and ECPS as a conditional test and QuickNavi™-Ebola RDT algorithm. We found cost-effectiveness in 88.2% of simulations for ECPS at -3 points of cut-off, 99.6% of simulations for the dual WHO case definition/QuickNavi™-Ebola RDT algorithm, and 88.9% of simulations for EPCS at -3 points of cut-off/QuickNavi™-Ebola RDT algorithm (**Fig 8**).

The dual screening algorithm with ECPS as a conditional test with QuickNavi™-Ebola RDT was cost-effective compared to any other screening algorithm at the country-specific WTP threshold. All algorithms were cost-effective in 100% of simulations if the WTP threshold of 3 times GDP per capita per isolated EVD is used, except algorithms 2, 5, and 6 (**S5 Fig**).

## Discussion

Current observations demonstrate that the performance accuracy of the WHO case definition for EVD suspect cases is inadequate due to its low sensitivity and specificity. Therefore, its use in the screening of EVD-suspicious cases leads to suboptimal effectiveness in the isolation process during outbreaks [14–16].

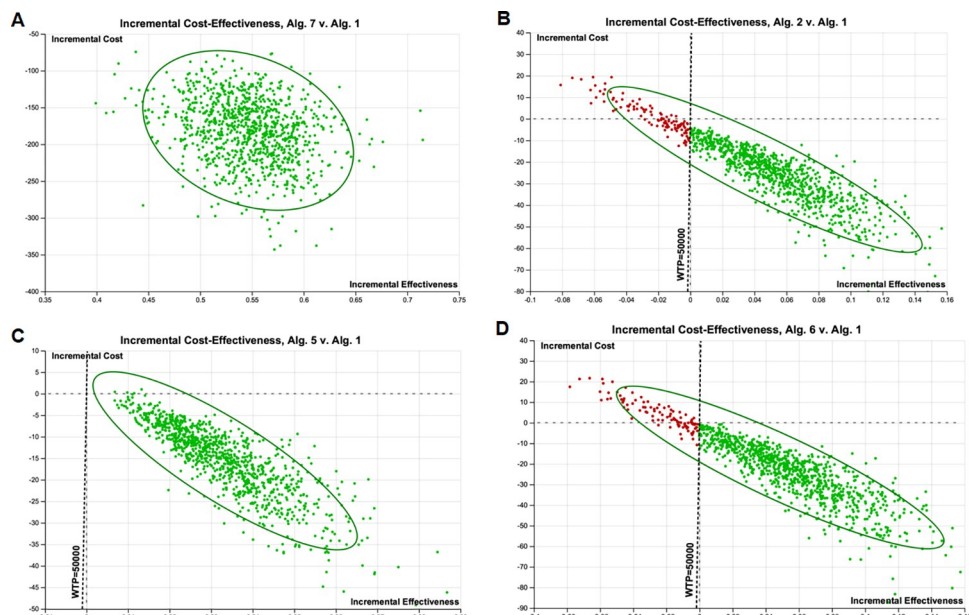

**Fig 7. Incremental cost-effectiveness of each algorithm compared to the WHO case definition-screening algorithm (Algorithm 1) during iterations of Monte Carlo simulation.** The ellipse represents 95% confidence points. The diagonal dashed line represents ICERs at a WTP threshold of USD 50 000. Points to the right of this dashed line are considered cost-effective. Dotted horizontal line shows incremental cost of USD 0. Points below this line represent iterations in which the given algorithm was cost- saving in 100% of simulation compared to Algorithm 1. This figure does not present all simulations of algorithms compared to algorithm 1. Those figure not presented here were cost- saving in 100% of simulation compared to algorithm 1 at this WTP threshold. Green points: ICERs that fall below the WTP line in Monte Carlo simulations, the maximum acceptable ICER (the algorithm is considered cost-effective); Red points: ICERs that fall above the WTP line, the maximum acceptable ICER (the algorithm is considered costly and less effective). Abbreviations: Alg. = algorithm; EVD = Ebola virus disease; ECPS = extended clinical prediction score; WTP = willingness-to-pay; ICER = incremental cost-effectiveness ratio.

However, during EVD outbreaks, health professionals use the clinical criteria to isolate suspected Ebola cases while they await confirmation by the GeneXpert® test. Our study findings show that incorporating scoring and RDT tools into the screening algorithms for suspect cases improves the efficiency of isolating EVD suspect cases. The WHO case definition algorithm used is less effective and costly than the other screening algorithms evaluated.

The ECPS as a joint or conditional algorithm and the dual screening algorithms (which combine the ECPS as a joint or conditional test with the QuickNavi™-Ebola RDT algorithms) provided the highest number of EVD cases (true positives) isolated for cost in our findings. From a health system perspective, our analysis shows that incorporating screening with ECPS as a conditional test algorithm and dual algorithm testing with ECPS as a conditional test and QuickNavi™-Ebola RDT into EVD case finding was highly cost-effective. Thus, these algorithms were inexpensive, more effective, and cost—saving compared to the current WHO case definition algorithm or any other competing algorithms.

Moreover, our analysis showed that, in the context of the low cost of SOC, the high cost of QuickNavi™-Ebola RDT resulted in changes in the ranking of algorithm efficiencies, and ECPS as a conditional became the most cost-effective. Therefore, choosing an algorithm will depend on the cost of both SOC per course and QuickNavi™-Ebola RDT. The prevalence of EVD in outbreaks ranges between 2 and 10% in the suspected population [1, 26]. Our conclusions, e.g., the ranking of algorithms, changed due to the variation in prevalence in this range in the one-way sensitivity analysis. In addition, the findings showed that the variations in the sensitivity of the QuickNavi™-Ebola RDT resulted in changes in the algorithms' efficiency ranking.

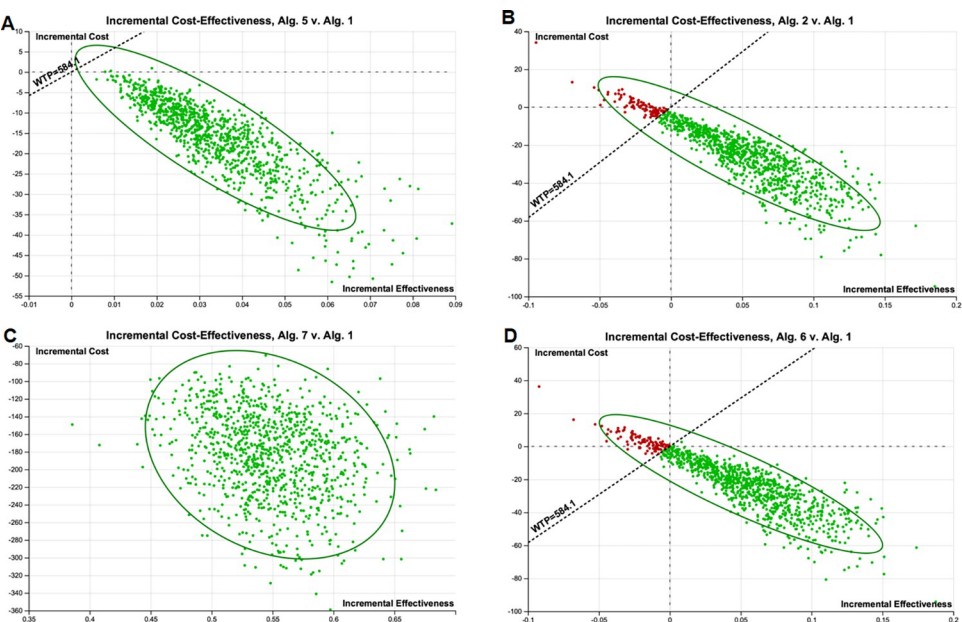

**Fig 8. Incremental cost-effectiveness of each algorithm compared to WHO case definition algorithm (Algorithm 1) during 1000 iterations of Monte Carlo simulation at a WTP threshold of USD 584.1.** The ellipse represents 95% confidence points. The diagonal dashed line represents ICERs at a WTP threshold of USD 584.1. Points to the right of this dashed line are considered cost-effective. Dotted horizontal line shows incremental cost of USD 0. Points below this line represent iterations in which the given algorithm was cost- saving compared to Algorithm 1. This figure does not present all simulations of algorithms compared to algorithm 1. Those not presented here were cost- saving in 100% of simulation compared to algorithm 1 at this WTP threshold. Green points: ICERs that fall below the WTP line in Monte Carlo simulations, the maximum acceptable ICER (the algorithm is considered cost-effective); Red points: ICERs that fall above the WTP line, the maximum acceptable ICER (the algorithm is considered costly and less effective). Abbreviations: Alg. = algorithm; WTP = willingness-to-pay; ICER = incremental cost-effectiveness ratio.

We did not observe any variation in any other sensitivities or specificities of the tests used in the model that would influence the algorithm's ranking. However, the variation in the SOC cost changed the algorithms ranking at the lower estimates, i.e., USD 150 per course. No marketed price for the QuickNavi™-Ebola RDT exists as of now [38]. If the cost of the SOC is less than 150 USD and the estimated price of the RDT is more than 10 USD per test, using the ECPS as a joint or conditional test for screening will provide better value for money (cost) for the overall health gained.

According to the results of our model, screening the ECPS as a conditional test algorithm and with the dual ECPS as a conditional test with the QuickNavi™-Ebola RDT algorithm resulted in USD 88.6 and 91.4 of ICER per additional EVD case isolated, respectively. Additionally, it could be considered cost-effective when a USD 50 000 WTP threshold was applied. From the perspective of the DRC public health system, the dual ECPS as a conditional test with the QuickNavi™-Ebola RDT algorithm can be considered suitable for screening of EVD suspect cases in outbreak settings. When the scenario of the DRC per capita domestic annual gross product is applied to our analysis, the cost of screening with dual ECPS as a conditional test and the QuickNavi™-Ebola RDT algorithm would be above the WTP threshold (USD 584.1 per additional patient correctly classified).

The WHO recommendations consider that interventions costing more than three times the gross domestic product are not cost-effective [42]. In our findings, the cost of ECPS as a joint test, ECPS as a conditional test, and dual ECPS as a joint test with QuickNavi™-Ebola RDT algorithms did not exceed the WHO-CHOICE threshold. Therefore, we observed that

although these three algorithms mentioned above did not find highly cost-effective alternatives, they met the DRC public health system's perspective.

At baseline estimates, our results showed that the difference between the cost-effectiveness of the dual algorithm (i.e., ECPS as a conditional test associated with the QuickNavi™-Ebola RDT) and ECPS as a joint or conditional test algorithm was marginal. In a context associated with the scarcity of Ebola RDTs on the market or the difficulty of scaling them up, introducing ECPS as a joint or conditional test for screening could be a better choice. Indeed, none of our algorithm rankings changed in 100% of the simulations when we looked at the 2021-DRC-GDP per capita willing-to-pay threshold compared with the WHO case definition. All other factors are equal, the cost-effectiveness threshold is the amount a decision-maker is willing-to-pay for a unit of the health effect. Therefore, the cost-effect analysis should be based on specific health effect targets to achieve, specific budget constraints to keep in mind, and the willing-to-pay ceiling provided by the primary user of the analysis results. In this way, the objective of the analysis could be to minimize the cost for the health effect target, maximize the health effect for the budget constraint, or know what algorithm to consider cost-effective. A cost-effectiveness threshold, such as the per capita gross product, is usually chosen to identify the screening algorithms that provide the best value for the cost. Alternatively, whether that threshold represents a better reference for countries, willingness-to-pay remains unclear, as no evidence of a linear relationship was established between them. The per capita gross product usually does not constitute the social willingness-to-pay. This last includes not only the market willingness-to-pay but also the nonmarket values, i.e., social preferences. Thus, the choice of the threshold depends on how decision-makers, health managers, and patients weigh the value of health benefits. Patients or healthcare managers could use other preferences, resulting in an overestimation of the value of the health benefit and leading to a very stringent threshold that can rule out some efficient algorithms. Conversely, the deciders not directly concerned by the given health problem (those living far from the epicenter of the outbreaks) could value the health benefit differently with a lax threshold, resulting in the inclusion of some inefficient options. From the healthcare perspective, choosing the cost-effectiveness threshold could lead to an important opportunity cost for the providers, e.g., healthcare workers directly concerned with managing scarcely available resources to gain health.

Therefore, choosing a threshold to identify cost-effective algorithms to implement must mean reaching a real consensus that places thresholds (ICERs) in the context of their application (a choice that considers local policies and managerial options such as funding resources, ethics, feasibility, local participation, etc.) [43–45]. Moreover, findings from this study support the idea that it is worth using some algorithms to screen EVD suspects in outbreak contexts with available emergency funding during the epidemic period. However, integrating these cost-effective algorithms into the Ebola surveillance system requires additional analysis, including a budget impact and feasibility assessment. The budget impact analysis will assess whether the adoption of a new EVD screening strategy is affordable. This will help quantify the financial impact of the adoption, given the resource and budget constraints in low- or middle-income countries and the number of unmet needs for the budget holder (i.e., the health system, government, etc.) [46].

This study explicitly responds to the World Health Organization's call for an innovative EVD screening strategy, as it assesses the cost-effectiveness aspects and provides valuable data for decision-makers in the context of increased EVD outbreaks in countries in Central and West Africa. However, no commercial RDT is available, and our study used only one RDT in its analyses. Including more than one RDT would give better insight into which RDT to use in screening. No algorithm built into the model was evaluated prospectively. Thus, different algorithms should be evaluated in future outbreaks to assess their real impact.

## Conclusion

This study demonstrates that in screening EVD suspects, Ebola clinical prediction scores as rapid diagnostic tools and QuickNavi™-Ebola RDT can be highly cost-effective compared with the traditional WHO clinical case definition.

If prediction scores and RDT are adopted, using dual ECPS as a conditional test with the QuickNavi™-Ebola RDT algorithm is the best screening option as it lowers the costs of confirmation testing and overall care during an EBOV epidemic. In some circumstances, such as those with a low cost of SOC, using the ECPS as a joint or conditional test to screen EVD suspects could be cost-effective in the DRC context. However, additional analyses that investigate the affordability and feasibility and account for all stakeholders' preferences are required to support their extended use in the surveillance system for the Ebola virus disease in the concerned countries.

## Supporting information

**S1 File. The detailed description of algorithms tested in the decision tree model.**
(DOCX)

**S2 File. The supplementary appendix.**
(DOCX)

**S3 File. CHEERS checklist.**
(DOCX)

**S1 Table. Cost-effectiveness ratios (USD per EVD isolated) in relation to variation in the cost of QuickNavi™-Ebola RDT and cost of standard-of-care (SOC).**
(DOCX)

**S1 Fig. Complete decision tree model.**
(TIF)

**S2 Fig. Variations in cost-effectiveness ratios of eight Ebola screening algorithms as a function of sensitivities of the WHO case definition for the suspect and ECPS at -3 points of cut-off. A** is the effect of variation in the sensitivity of the WHO case definition for the suspect on the efficiency of algorithms. **B** is the effect of variation in the sensitivity of the ECPS at -3 points of cut-off on the efficiency of algorithms.
(TIF)

**S3 Fig. Variations in cost effectiveness ratios of the eight Ebola screening algorithms as a function of the cost of standard-of-care and QuickNavi™-Ebola RDT. A** presents the effect of variation in the cost of standard-of-care on the efficiency of the eight Ebola screening algorithms. **B** presents the effect of variation in the QuickNavi™-Ebola RDT cost on the efficiency of the 8 Ebola screening algorithms.
(TIF)

**S4 Fig. Variations in the effectiveness and cost of the eight Ebola screening algorithms as a function of the prevalence of Ebola virus disease in the suspected population. A** depicts the effect of variation in the prevalence of Ebola virus disease on the effectiveness of screening algorithms. **B**, the effect of variation in the prevalence of Ebola virus on the cost of screening algorithms. The dotted horizontal line shows the threshold value of the prevalence over which the cost of the algorithm changes. Over this threshold of 10% of disease prevalence, the cost of ECPS as a joint or conditional test becomes low. Abbreviations: Alg. = algorithm;

ECPS = extended clinical prediction score; EVD = Ebola virus disease.
(TIF)

**S5 Fig. Incremental cost-effectiveness of each algorithm compared to WHO case definition- algorithm (Algorithm 1) during 1000-iterations of Monte Carlo simulation at a WTP threshold of USD 1, 752.3.** The ellipse represents 95% confidence points. The diagonal dashed line represents ICERs at a WTP threshold of USD 1,752.3. Points to the right of this dashed line are considered cost-effective. The dotted horizontal line shows an incremental cost of USD 0. Points below this line represent iterations in which an algorithm was cost saving compared with algorithm 1. This figure does not present all simulations of algorithms compared to algorithm 1. Those not presented here were cost- saving in 100% of simulations compared to algorithm 1 at this WTP threshold. Green points: ICERs that fall below the WTP line in Monte Carlo simulations, the maximum acceptable ICER (the algorithm is considered cost-effective); Red points: ICERs that fall above the WTP line, the maximum acceptable ICER (the algorithm is considered costly and less effective). Abbreviations: Alg. = algorithm; WTP = willingness to pay; ICER = incremental cost-effectiveness ratio.
(TIF)

**S1 Data.**
(ZIP)

## Acknowledgments

We are grateful to Professor Lutgarde Lynen from the Institute of Tropical Medicine in Antwerp, Belgium, for her helpful comments on the manuscript.

## Author Contributions

**Conceptualization:** Antoine Oloma Tshomba, Charles T. Kayembe, Faustin M. Chenge, Dieudonné N. Mumba, Désiré D. Tshala-Katumbay, Sabue Mulangu.

**Data curation:** Antoine Oloma Tshomba, Daniel Mukadi-Bamuleka, Olivier M. Tshiani.

**Formal analysis:** Antoine Oloma Tshomba, Daniel Mukadi-Bamuleka, Faustin M. Chenge.

**Funding acquisition:** Placide Mbala-Kingebeni, Steve Ahuka-Mundeke, Désiré D. Tshala-Katumbay, Sabue Mulangu.

**Investigation:** Antoine Oloma Tshomba, Daniel Mukadi-Bamuleka, Sabue Mulangu.

**Methodology:** Antoine Oloma Tshomba, Anja De Weggheleire, Charles T. Kayembe, Bart Karl M. Jacobs, Dieudonné N. Mumba, Désiré D. Tshala-Katumbay, Sabue Mulangu.

**Project administration:** Jean-Jacques Muyembe-Tamfum, Dieudonné N. Mumba, Désiré D. Tshala-Katumbay, Sabue Mulangu.

**Supervision:** Charles T. Kayembe, Jean-Jacques Muyembe-Tamfum, Dieudonné N. Mumba, Désiré D. Tshala-Katumbay, Sabue Mulangu.

**Validation:** Daniel Mukadi-Bamuleka, Olivier M. Tshiani, Placide Mbala-Kingebeni, Sabue Mulangu.

**Visualization:** Antoine Oloma Tshomba.

**Writing – original draft:** Antoine Oloma Tshomba, Anja De Weggheleire.

**Writing – review & editing:** Antoine Oloma Tshomba, Daniel Mukadi-Bamuleka, Anja De
Weggheleire, Olivier M. Tshiani, Charles T. Kayembe, Placide Mbala-Kingebeni, Jean-
Jacques Muyembe-Tamfum, Steve Ahuka-Mundeke, Faustin M. Chenge, Bart Karl M.
Jacobs, Dieudonné N. Mumba, Désiré D. Tshala-Katumbay, Sabue Mulangu.

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
