## [Decision Letter · Decision Letter 0]

23 May 2023

PONE-D-23-12291Cost-effectiveness of incorporating Ebola prediction score tools and rapid diagnostic tests into a screening algorithm: a decision analytic modelPLOS ONE

Dear Dr. Tshomba,

Thank you for submitting your manuscript to PLOS ONE. After careful consideration, we feel that it has merit but does not fully meet PLOS ONE’s publication criteria as it currently stands. Therefore, we invite you to submit a revised version of the manuscript that addresses the points raised during the review process.

We look forward to receiving your revised manuscript.

Kind regards,

Jan Rychtář

Academic Editor

PLOS ONE

Journal Requirements:

Additional Editor Comments:

The reviewer makes several suggestions for revisions and improvements and I urge the authors to take all of these suggestions into considerations and to make appropriate changes to their manuscript

Reviewers' comments:

Reviewer's Responses to Questions

**Comments to the Author**

1. Is the manuscript technically sound, and do the data support the conclusions?

Reviewer #1: Yes

2. Has the statistical analysis been performed appropriately and rigorously? 

Reviewer #1: Yes

3. Have the authors made all data underlying the findings in their manuscript fully available?

Reviewer #1: No

4. Is the manuscript presented in an intelligible fashion and written in standard English?

Reviewer #1: Yes

5. Review Comments to the Author

Reviewer #1: Tshomba et al performed a computational decision tree analysis to compute the cost-effectiveness of different Ebola screening algorithms. They determined that combining the WHO case definition with the ECPS screening procedure and additionally screening negative suspect cases with the QuickNavi molecular rapid test is the most cost-effective strategy. Additional sensitivity analyses indicate that this recommendation holds across the range of plausible parameter measurement error. The approach used is sensible and is applied reasonably. However, I have a few technical questions/comments that are relevant to the overall conclusions; in particular, I wonder whether the cost of missing an infected individual during screening is appropriately estimated here. The manuscript’s findings are relevant to healthcare workers and public-health officials involved in Ebola outbreak response and address a very important issue with real-world implications. Therefore, I recommend publication after the technical issues have been addressed.

MAJOR COMMENTS:

- Should the potential for additional infections in the community be incorporated into the cost of missing a case (false negative error)? The payoff for this outcome is set to 0 “by assumption” in Table 3, which I’m guessing was chosen to simply represent the lack of benefit acquired from correctly isolating a case. However, missing a true Ebola infection may have additional public-health consequences, as the infected individual may infect several others in the community, which is not considered here. In the current analysis, the cost of erroneous isolation (false positive) is greater than the cost of missing a true case, which would not make sense if an infected individual allowed to remain in the community is expected to infect 2 or more others. Please consider incorporating these secondary effects into your analysis, as the risk of infecting others due to a false negative screening result may outweigh the risk of erroneous isolation, which is quite costly in the current analysis.

- Can the cost of erroneously isolating an uninfected individual (given in Table 3) be reduced by improving isolation practices to reduce the probability of becoming infected as a result of isolation?

- It is not clear to me what the precise definitions of screening algorithms 3 and 4 are. They are simply described as “2) ECPS as a join [sic] test, 3) ECPS as a conditional test” (lines 144-145). The tree in Fig. 1 does not clarify this point, but rather simply labels a branch as “ECPS as a conditional test”. From that description, I’m not sure what other test(s) are performed in addition to the ECPS. Since the molecular RDT is described separately and the only other screening algorithm mentioned is the WHO criteria, I assume the ECPS is being used joint/conditionally with the WHO criteria. Also, in terms of the conditional version of the test, I’m not sure which screening protocol is being applied first. Please clarify further in the text.

- The results in Tables 4 and 5 for algorithms 3 and 4 are exactly the same, which seems unlikely given their different sensitivities/specificities in Table 2. Is this expected?

- I’m confused as to why the sensitivity analyses in Fig. 2B-C show no change in cost-effectiveness for any condition tested. For example, a back-of-the-envelope calculation suggests that in 2C, if the sensitivity of the conditional test increases by ~12% (as it does over the range of the x-axis, 0.61 to 0.69), then the cost-effectiveness ratio of algorithm 4 should decrease substantially, from ~$106/case to ~$95/case, since the number of cases identified should go up by 12% as well (from Equation 4 in S1 Text). This change should be visible in Fig. 2C but is not. Please explain.

- The sensitivities of the ECPS joint and conditional tests (algorithms 3 and 4) are presumably not independent variables, since both are functions of the sensitivities of the ECPS and the other screening method with which the ECPS was combined (again, possibly the WHO definition?). Therefore, if the joint sensitivity changes, the conditional sensitivity will also change, and treating these two values as separate quantities in the sensitivity analysis doesn’t make much sense. A better choice might be to perform the sensitivity analysis in Fig. 2 on the sensitivities of the individual tests making up the joint/conditional screening protocols (i.e., the ECPS and WHO protocol sensitivities).

- Is the WTP value of $50,000 used in Fig. 6 and specified as the “default” value in the Methods appropriate in any circumstance? It seems to me that the country specific WTP threshold (used in Fig. 7, for example) is much more relevant and that the $50,000 value was chosen arbitrarily and is far too high for the likely use contexts of these screening algorithms.

MINOR COMMENTS:

- The description of algorithm 4 in Table 1 should indicate that this is a conditional test, not a joint test.

- “Joint test” is sometimes written erroneously as “join test”.

- The resolution of some of the original figure files (e.g., Fig. 2) are quite low, making them difficult to read.

- A couple of clauses end with “or so” (e.g., lines 536, 545), which I don’t believe is standard English- perhaps they can be replaced with “etc”?

- Perhaps defining the term “incremental” (e.g., cost vs. incremental cost) in the text would help readers who are unfamiliar with formal cost-effectiveness analysis.

- Table 5 includes a lot of numerical data and is difficult to interpret quickly. Perhaps plotting these data as a heatmap, for example, would make them more interpretable? The raw tabular data can be included in the supplement.

- Raw numerical outputs from key parts of the cost-effectiveness analysis are included in the main text (Tables 4 and 5). However, raw data from most of the sensitivity analyses (Figs. 3-7) are not available. These data are probably reproducible using the commercial software used by the authors, but this would not be accessible to many readers.

6. PLOS authors have the option to publish the peer review history of their article (what does this mean?). If published, this will include your full peer review and any attached files.

Reviewer #1: **Yes: **Debra Van Egeren

---

## [Author Response · Author response to Decision Letter 0]

20 Jun 2023

Kinshasa, Jun 20, 2023

Tshomba Oloma Antoine

Institut National de Recherche Biomédicale (INRB) 

Kinshasa, Dem. Rep. of Congo, 

antotshomba@yahoo.fr

+243 815602451

Jun 20, 2023

Jan Rychtář

Academic Editor

PLOS ONE Journal

plosone@plos.org

Dear Editor,

We are resubmitting our manuscript entitled “Cost-effectiveness of incorporating Ebola prediction score tools and rapid diagnostic tests into a screening algorithm: a decision analytic model” as a Research Article for consideration of publication in the PLOS ONE Journal. 

First, we want to express our gratefulness to the Editor and Reviewers’ for their overall very positive comments on our work and their suggestions for improvement. Through this letter, we have tried answering to the best of our knowledge the questions, suggestions and remarks provided by the Editors and Reviewers. 

None of the authors has a competing interest to declare and our manuscript has not been submitted, or accepted elsewhere. All authors have contributed to, seen, and approved the final, submitted version of the manuscript.

We have upload the following documents:

- The clean version of the Manuscript, file labeled "Manuscript"

- The track changes version of the manuscript, file labeled “Revised Manuscript with Track Changes”. 

- The covering letter addressing the editorial and referees’ comments, file labeled “Response to Reviewers”.

- We made available, as raw data, outputs from our main analyses, e.g., one-way sensitivity analysis outputs and outputs from probabilistic sensitivity analysis.

- This economic study did not constitute human research. Thus, participant consent was not applicable, and the study did not require ethical approval. 

- At last, after creating and adding other figures and supplementary files, we re-ordered them in the text.

Again thank you for all deep and helpful remarks and recommendations you have addressed on our manuscript. 

We will look forward to hearing whether this manuscript can be considered of interest for publication in the PLOS ONE Journal and remain at your disposal for any required clarifications. 

Yours sincerely,

Antoine Tshomba

PONE-D-23-12291

Cost-effectiveness of incorporating Ebola prediction score tools and rapid diagnostic tests into a screening algorithm: a decision analytic model

PLOS ONE

Dear Dr. Tshomba,

Thank you for submitting your manuscript to PLOS ONE. After careful consideration, we feel that it has merit but does not fully meet PLOS ONE’s publication criteria as it currently stands. Therefore, we invite you to submit a revised version of the manuscript that addresses the points raised during the review process.

We look forward to receiving your revised manuscript.

Kind regards,

Jan Rychtář

Academic Editor

PLOS ONE

Journal Requirements:

Thank you very much for providing to me these PLOS ONE formatting guidelines. I am using them to correct and adapt to PLOS ONE requirements

Thank you very much for your advice 

This economic study did not constitute human research. Thus, participant consent were not applicable and the study did not require ethical approval.

We provided this statement in the manuscript, on lines 308 to 311 in Ethic statements point of the methods

Our study did not constitute human research. It was part of the Ebola outbreak response and disease surveillance in the North-Kivu Ebola outbreak DRC. This economic evaluation study used published results from the literature to build the decision model. Thus, it did not require ethical approval.

Thank you Dear Editor for your advice. 

We provided minimal analyses’ outputs and added this statement in Data availability statements: lines 629 to 632

This modeling study was an economic evaluation that built its model using publicly accessible information from the literature. We made available, for transparency purposes, the output reports of our analyses, e.g., the one-way and probabilistic sensitivity analysis outputs.

Additional Editor Comments:

The reviewer makes several suggestions for revisions and improvements and I urge the authors to take all of these suggestions into considerations and to make appropriate changes to their manuscript

Thank you Dear Editor.

Indeed, all these suggestions were really helpful and fundamental for our manuscript's improvement. We tried to take, at our best, all these suggestions into consideration.

Reviewers' comments:

Reviewer's Responses to Questions

Comments to the Author

1. Is the manuscript technically sound, and do the data support the conclusions?

Reviewer #1: Yes

2. Has the statistical analysis been performed appropriately and rigorously?

Reviewer #1: Yes

3. Have the authors made all data underlying the findings in their manuscript fully available?

Reviewer #1: No

4. Is the manuscript presented in an intelligible fashion and written in standard English?

Reviewer #1: Yes

5. Review Comments to the Author

Reviewer #1: Tshomba et al performed a computational decision tree analysis to compute the cost-effectiveness of different Ebola screening algorithms. They determined that combining the WHO case definition with the ECPS screening procedure and additionally screening negative suspect cases with the QuickNavi molecular rapid test is the most cost-effective strategy. Additional sensitivity analyses indicate that this recommendation holds across the range of plausible parameter measurement error. The approach used is sensible and is applied reasonably. However, I have a few technical questions/comments that are relevant to the overall conclusions; in particular, I wonder whether the cost of missing an infected individual during screening is appropriately estimated here. The manuscript’s findings are relevant to healthcare workers and public-health officials involved in Ebola outbreak response and address a very important issue with real-world implications. Therefore, I recommend publication after the technical issues have been addressed.

We appreciate you making this observation.

We updated and incorporated the cost of missing a real Ebola case into our model. We have modified our conclusion to reflect that the cost of incorrectly ruling out an Ebola case includes or reflects the cost of managing the patient who was incorrectly ruled out as well as the cost of any potential Ebola cases that this false-negative case could lead to in the community.

Our supplemental appendix contains a corrected version of the cost formula.

MAJOR COMMENTS:

- Should the potential for additional infections in the community be incorporated into the cost of missing a case (false negative error)? The payoff for this outcome is set to 0 “by assumption” in Table 3, which I’m guessing was chosen to simply represent the lack of benefit acquired from correctly isolating a case. However, missing a true Ebola infection may have additional public-health consequences, as the infected individual may infect several others in the community, which is not considered here. In the current analysis, the cost of erroneous isolation (false positive) is greater than the cost of missing a true case, which would not make sense if an infected individual allowed to remain in the community is expected to infect 2 or more others. Please consider incorporating these secondary effects into your analysis, as the risk of infecting others due to a false negative screening result may outweigh the risk of erroneous isolation, which is quite costly in the current analysis.

Thank you for bringing this to my attention. You are totally right!

Indeed, no risk zero exists for a deadly infection like Ebola. Thus, we agreed with you that not incorporating this in our analysis could constitute a big limitation of the study.

In light of the findings of these studies, by Leroy et al. and Mbala et al., which suggest that the role of these false-negative cases (composed of asymptomatic and mildly symptomatic patients) in human-to-human disease transmission in the community is poorly understood and, if there is one, this role is very low, we assumed a reduction of disease secondary attack rate up to 50% and assigned a false-negative patient erroneously ruled out less the likelihood that these false-negative cases generate Ebola cases in the community and computed its payoff as described in supplementary appendix

 (References to include: Leroy et al., Early immune responses accompanying human asymptomatic Ebola infections) Clin Exp Immunol 2001, Vol. 124, Issue 3, Pages 453–60, and Mbala et al. Evaluating the frequency of asymptomatic Ebola virus infection (Philos Trans R Soc Lond B Biol Sci 2017 Vol. 372 Issue 1721)

Therefore, we added these statements:

In method part of the manuscript: lines 214 to 224

The effectiveness reward for erroneous negatives was calculated in the same manner as for false positives. We used the method described in the supplementary appendix and referenced the findings by Leroy et al. and Mbala et al. [ref. to include] to make our assumptions. We assumed a 50% reduction in the disease secondary attack rate for the transmission in the community by these false negative patients and assigned a false negative patient erroneously ruled out minus the random probability that these false negative cases generate Ebola cases in the community. Therefore, for each EVD instance that was ruled out, we assigned a value of -0.00125 as the effectiveness payoff.

In the results part of the manuscript:

We updated Tables 4 and 5, Figures 4 to 7, and Supplementary Figures.

In the text, we updated where needed accordingly.

- Can the cost of erroneously isolating an uninfected individual (given in Table 3) be reduced by improving isolation practices to reduce the probability of becoming infected as a result of isolation?

We appreciate you sharing this, and we think the concept is appealing.

At the screening stage, while awaiting the lab confirmation results, all isolated suspects receive the same intervention, such as standard of care, test confirmation, and other associated interventions. Therefore, at this point, the secondary attack rate (SAR) change has no longer had an impact on the price of FP. The specificity of the screening test being performed will largely determine how the FP cost will change at this step. A SAR reduction, however, could, following confirmation, lessen the amount of iatrogenic EVDs created in isolation wards, thereby lowering the cost of FP care with expensive EVD-specific treatment.

We concentrated on recording all costs incurred during the screening procedure for this study; hence, we simply recorded the overall cost or expenses.

To understand how a drop in the disease SAR would affect the cost of FP isolation, we manually computed this cost, assuming any cost for EVD-specific treatment.

Costs due to FP at the SAR of 2.5% would be reduced by 20% at the SAR of 2% and by 60% at the SAR of 1%. 

However, this was not the purpose of our analysis.

- It is not clear to me what the precise definitions of screening algorithms 3 and 4 are. They are simply described as “2) ECPS as a join [sic] test, 3) ECPS as a conditional test” (lines 144-145). The tree in Fig. 1 does not clarify this point, but rather simply labels a branch as “ECPS as a conditional test”. From that description, I’m not sure what other test(s) are performed in addition to the ECPS. Since the molecular RDT is described separately and the only other screening algorithm mentioned is the WHO criteria, I assume the ECPS is being used joint/conditionally with the WHO criteria. Also, in terms of the conditional version of the test, I’m not sure which screening protocol is being applied first. Please clarify further in the text.

We sincerely appreciate it.

In addition to fixing the error in Table 1, we also provided a supplementary that details each algorithm definition (S1 File) and a plot depicts the entire decision tree model's graph (S1 Fig.).

We added this statement:

On lines 160 and 162: 

Supplementary figure S1 draws the complete decision tree model, and supplementary file S1 describes and defines each algorithm tested in the model (S1 Fig. and S1 File).

On lines 170 to 175 of the procedure part's legend for Fig. 1, we added the following statement:

Algorithms 1, 2, and 3 use a single screening test; their visual representations are similar to the branch shown on the decision branch of algorithm 4. Algorithms 6, 7, and 8 use two sequencing screening tests; their visual presentations look like this, shown in this format on algorithm 5's branch. QuickNavi™-Ebola RDT is used after the first screening test in algorithms with two screening tests.

We also updated Supplementary figures and file numeration

- The results in Tables 4 and 5 for algorithms 3 and 4 are exactly the same, which seems unlikely given their different sensitivities/specificities in Table 2. Is this expected?

We are grateful for your observation.

We thoroughly rechecked our model and its inputs, and we found a problem in the two branches because they had the same sensitivity and specificity inputs.

We corrected them and revised Tables 4 and 5 and the text with the required modifications.

In addition, we added the cost of an EVD case that was mistakenly ruled out (a false negative) and updated our manuscript where needed.

- I’m confused as to why the sensitivity analyses in Fig. 2B-C show no change in cost-effectiveness for any condition tested. For example, a back-of-the-envelope calculation suggests that in 2C, if the sensitivity of the conditional test increases by ~12% (as it does over the range of the x-axis, 0.61 to 0.69), then the cost-effectiveness ratio of algorithm 4 should decrease substantially, from ~$106/case to ~$95/case, since the number of cases identified should go up by 12% as well (from Equation 4 in S1 Text). This change should be visible in Fig. 2C but is not. Please explain.

We are very grateful for you.

After double-checking our model and its inputs, we changed the corresponding graph; however, the results under examination in the interval range on which we applied our one-way sensitivity analysis exhibited no change. 

We were aware that their variation had little impact on the cost-effectiveness ratio within the range of the believable interval in which we did our sensitivity analysis. 

We also understood a nonlinear relationship between the prevalence of diseases and outcomes, such as the cost-effectiveness ratio. 

The calculation is performed at the expected value at the root of each algorithm. This takes into account every possible result in this decision branch, which is reliant on several other probabilities in the branch. The modification is not remarkable as a result of this. 

However, we attempted to explain the observed tendencies using particular data from the analyses' raw outputs, where needed.

- The sensitivities of the ECPS joint and conditional tests (algorithms 3 and 4) are presumably not independent variables, since both are functions of the sensitivities of the ECPS and the other screening method with which the ECPS was combined (again, possibly the WHO definition?). Therefore, if the joint sensitivity changes, the conditional sensitivity will also change, and treating these two values as separate quantities in the sensitivity analysis doesn’t make much sense. A better choice might be to perform the sensitivity analysis in Fig. 2 on the sensitivities of the individual tests making up the joint/conditional screening protocols (i.e., the ECPS and WHO protocol sensitivities).

Again thank you for your advice.

We also performed sensitivity analyses on the ECPS and WHO case definition sensitivities.

We also provided as a supplementary Figure S2, a figure that plots one-way sensitivity analysis investigating the uncertainty of outcomes due to variation of ECPS and WHO case definition sensitivities.

- Is the WTP value of $50,000 used in Fig. 6 and specified as the “default” value in the Methods appropriate in any circumstance? It seems to me that the country specific WTP threshold (used in Fig. 7, for example) is much more relevant and that the $50,000 value was chosen arbitrarily and is far too high for the likely use contexts of these screening algorithms.

Thank you very much for this observation.

In fact, after re-checking our model inputs and including the cost of erroneously ruling out false negative cases, this WTP appeared to be relevant. As recommended, using this $50,000 WTP allows the results to be interpreted in various global contexts.

Thus, we updated the current Fig. 6 and its numeration, after adding one more figure in the manuscript.

MINOR COMMENTS:

- The description of algorithm 4 in Table 1 should indicate that this is a conditional test, not a joint test.

Thank you very much. We corrected the error in the Table 1

- “Joint test” is sometimes written erroneously as “join test”.

Thank you very much. We corrected this in the Table 

- The resolution of some of the original figure files (e.g., Fig. 2) are quite low, making them difficult to read.

Thank you very much. 

We corrected by improving the resolution. Thank you.

- A couple of clauses end with “or so” (e.g., lines 536, 545), which I don’t believe is standard English- perhaps they can be replaced with “etc”?

Thank you very much. 

We corrected them. Thank you.

- Perhaps defining the term “incremental” (e.g., cost vs. incremental cost) in the text would help readers who are unfamiliar with formal cost-effectiveness analysis.

Thank you for this recommendation

We added this on lines 266 to 271

Where the numerator, in the case of Ebola disease, represents the incremental cost, which is the total expense incurred due to an additional health effect, e.g., an isolated EVD case. It is calculated by looking at the additional expenses made throughout the screening process, such as supplies used, for one extra health effect. The denominator represents the incremental effectiveness, which is the increase in the effectiveness of the Ebola screening throughout the screening process.

- Table 5 includes a lot of numerical data and is difficult to interpret quickly. Perhaps plotting these data as a heatmap, for example, would make them more interpretable? The raw tabular data can be included in the supplement.

You are welcome for pointing this out.

As no marketed Ebola RDTs or treatments are available, we performed a two-way analysis with both the cost of QuickNavi-Ebola and the cost of SOC to capture these variations on algorithm ranking, thus the decision. We plotted a two-way sensitivity analysis and moved the relative Table 5 in supporting information as S1 Table 

We added these statements in method part of the manuscript on lines 283 to 288

Additionally, as there are currently no marketed RDTs or therapies for Ebola (they are still in negotiations), we performed a 2-way sensitivity analysis exploring the effects of changing the price of the QuickNavi-Ebola RDT and the price of SOC on algorithm ranking. Three levels of 2017-DRC GDP, as a willingness to pay, were used in this analysis (at one, two, and three times the GDP).

At results part, we added updated the text on lines 406 to 409 and these statements to present the two-way figure.

Fig 4. Two-way sensitivity analysis comparing the net heath benefit of EVD screening algorithms 

Fig 4 legend:

A: at a willingness to pay of USD 584.1; B: at a willingness to pay of USD 1168.2; C: at a willingness to pay of USD 1752.3

 We added in supporting information

S1 Table. Cost-effectiveness ratios (USD per EVD isolated) in relation to variation the cost of QuickNavi™-Ebola RDT and cost of standard-of-care (SOC)

- Raw numerical outputs from key parts of the cost-effectiveness analysis are included in the main text (Tables 4 and 5). However, raw data from most of the sensitivity analyses (Figs. 3-7) are not available. These data are probably reproducible using the commercial software used by the authors, but this would not be accessible to many readers.

Yes. We made available all analyses outputs as supporting data.

6. PLOS authors have the option to publish the peer review history of their article (what does this mean?). If published, this will include your full peer review and any attached files.

Do you want your identity to be public for this peer review? For information about this choice, including consent withdrawal, please see our Privacy Policy.

Reviewer #1: Yes: Debra Van Egeren

 Thank you. Yes, we uploaded them here.

---

## [Decision Letter · Decision Letter 1]

2 Jul 2023

PONE-D-23-12291R1Cost-effectiveness of incorporating Ebola prediction score tools and rapid diagnostic tests into a screening algorithm: a decision analytic modelPLOS ONE

Dear Dr. Tshomba,

Thank you for submitting your manuscript to PLOS ONE. After careful consideration, we feel that it has merit but does not fully meet PLOS ONE’s publication criteria as it currently stands. Therefore, we invite you to submit a revised version of the manuscript that addresses the points raised during the review process.

While the reviewer appreciate that most of the previous concerns were addressed, there are still several issues that need to be improved/addressed. Please revise your manuscript accordingly addressing all reviewer's comments.

We look forward to receiving your revised manuscript.

Kind regards,

Jan Rychtář

Academic Editor

PLOS ONE

Additional Editor Comments:

While the reviewer appreciates that most of the previous concerns were addressed, there are still several issues that need to be improved/addressed.

Reviewers' comments:

Reviewer's Responses to Questions

**Comments to the Author**

1. If the authors have adequately addressed your comments raised in a previous round of review and you feel that this manuscript is now acceptable for publication, you may indicate that here to bypass the “Comments to the Author” section, enter your conflict of interest statement in the “Confidential to Editor” section, and submit your "Accept" recommendation.

Reviewer #1: (No Response)

2. Is the manuscript technically sound, and do the data support the conclusions?

Reviewer #1: Partly

3. Has the statistical analysis been performed appropriately and rigorously? 

Reviewer #1: Yes

4. Have the authors made all data underlying the findings in their manuscript fully available?

Reviewer #1: Yes

5. Is the manuscript presented in an intelligible fashion and written in standard English?

Reviewer #1: Yes

6. Review Comments to the Author

Reviewer #1: Thank you for addressing the majority of my questions and comments. However, I still have concerns about how the costs of false positives and false negatives are being estimated, which were further exposed while I was reviewing the changes made during this revision. I think additional clarification and/or justification is required for the term of the form (1-(1-θ)δ) in equations 12-15 in S2 File, which estimates the “probability of infection given random contact with an EVD patient”. The secondary attack rate θ is estimated in the relevant reference (Okware et al) as the fraction of contacts of an EVD case that get infected. Given that definition, I don’t understand why the duration of infectiousness (1/δ) is being used here in this way. I suppose if you’re assuming that the risk to each suspected case entering isolation is equivalent to the risk of coming into contact with a single EVD patient for one day, and you are assuming the published secondary attack rate is being estimated from household contacts who are exposed continuously for the entire duration of infectiousness, there is some possible justification for this equation, which would need to be further explained in the manuscript. Unfortunately I believe that this risk estimate cannot be appropriately justified using this reasoning- it’s not clear from the reference for the secondary attack rate value exactly how a contact is defined, and there are published estimates for the household attack rate that are much higher than 2.5% (see the meta-analysis in Dean et al. Clin Infect Dis 2016 https://doi.org/10.1093/cid/ciw114 which includes the reference cited here). A better way to estimate infection risk for false positives would probably be to use results from a study specifically designed to measure the risk of nosocomial infection in patients erroneously isolated in EVD units (e.g., Arkell et al Tropical Medicine and International Health 2016

https://doi.org/10.1111/tmi.12802) which seem to estimate a higher risk of infection than calculated in this manuscript (though the absolute risk of exposure is still reasonably low).

The estimate of risk of community transmission from a false negative (which was calculated using the same reasoning) is even more problematic. Again, the assumption being made seems to be that the expected number of cases resulting from a true EVD case being allowed to remain in the community is equivalent to the risk of a single person coming into contact with that EVD patient for one day, except now the secondary attack rate is 50% lower than was estimated previously (presumably to account for lower risk from “asymptomatic” EVD patients). This seems to me to be quite an underestimation of the expected number of cases. First, I think these patients are not truly “asymptomatic” and instead may have nonspecific symptoms or be presymptomatic, so I don’t think there is good justification for the assumption that their infectiousness is lower (especially by an arbitrary 50%, which the authors do not justify). Also, I’m guessing that many of these patients will return to their homes after a negative screening, potentially exposing multiple people (possibly in higher-risk caregiving roles) over multiple days, making the expected number of new cases resulting from a single false negative to be much higher than the 0.00125 calculated here. Using the R0 value for EVD community transmission (while perhaps a bit of an overestimate) for the effect of a false negative is likely more correct, and more easily justified, and would result in a false negative penalty that is orders of magnitude higher than the value currently used here.

Since these false negative/positive risks are very important factors in the cost-effectiveness analysis (and thus the conclusions of the whole manuscript), they require suitable justification before I can recommend publication. I apologize for not noticing these issues in my first review.

I have a couple additional minor comments:

- I think Eq. 13 in S2 File is missing a 1-Prev term?

- Please explicitly state in the data availability statement that the raw data are available in the Supporting Outputs Data so readers know exactly where to find them.

- Figs. 7 and 8 legends: define red/green colors of points

- It’s still a little unclear in the text what “joint” and “conditional” mean when referring to the ECPS test. The authors define these terms very clearly in their previous publication (ref. [26] in this manuscript, relevant passage copied below), and I’d suggest adding a similar description here or at least referencing this publication at the point in the text where these algorithms are defined.

“Finally, we evaluated our prediction models according to two additional clinical practice approaches in healthcare settings: joint and conditional tests or approaches. In both approaches, the suspects with no reported risk of exposure would be considered to not have the disease and the clinical team would act accordingly. No additional action, e.g., isolation, would be required. In the joint approach, the clinical team should clinically examine all suspects at low-, intermediate-, and high-risk reported exposure and recommend for isolation only those with a predicted probability of EVD greater than 5% (the cut-off chosen to maximize sensitivity, about 90 percent, in disease adverse context). In the conditional approach, the clinical team should isolate all suspects with high-risk reported exposure irrespective of their predicted probability of the disease and then suspects at low and intermediate reported exposure having an EVD-predicted probability greater than 5%.”

7. PLOS authors have the option to publish the peer review history of their article (what does this mean?). If published, this will include your full peer review and any attached files.

Reviewer #1: **Yes: **Debra Van Egeren

---

## [Author Response · Author response to Decision Letter 1]

25 Jul 2023

Kinshasa, July 25, 2023

Tshomba Oloma Antoine

Institut National de Recherche Biomédicale (INRB) 

Kinshasa, Dem. Rep. of Congo, 

antotshomba@yahoo.fr

+243 815602451

July 25, 2023

Jan Rychtář

Academic Editor

PLOS ONE Journal

plosone@plos.org

Dear Editor,

We are resubmitting our manuscript entitled “Cost-effectiveness of incorporating Ebola prediction score tools and rapid diagnostic tests into a screening algorithm: a decision analytic model” as a Research Article for consideration of publication in the PLOS ONE Journal. 

First, we want to express our gratefulness to the Editor and Reviewers for their overall very positive comments on our work and their suggestions for improvement. Through this letter, we have tried answering to the best of our knowledge the questions, suggestions and remarks provided by the Editors and Reviewers. 

None of the authors has a competing interest to declare and our manuscript has not been submitted, or accepted elsewhere. All authors have contributed to, seen, and approved the final, submitted version of the manuscript.

We have upload the following documents:

- The clean version of the Manuscript, file labeled "Manuscript"

- The track changes version of the manuscript, file labeled “Revised Manuscript with Track Changes”. 

- The covering letter addressing the editorial and referees’ comments, file labeled “Response to Reviewers”.

- We used secondary attack rates from meta-analysis by Dean et al and made changes where needed in the manuscript text, figures and supporting materials and data 

Again thank you a million for all deepest and helpful remarks and recommendations you have addressed on our manuscript. 

We will look forward to hearing whether this manuscript can be considered of interest for publication in the PLOS ONE Journal and remain at your disposal for any required clarifications. 

Yours sincerely,

Antoine Tshomba

PONE-D-23-12291R1

Cost-effectiveness of incorporating Ebola prediction score tools and rapid diagnostic tests into a screening algorithm: a decision analytic model

PLOS ONE

Dear Dr. Tshomba,

Thank you for submitting your manuscript to PLOS ONE. After careful consideration, we feel that it has merit but does not fully meet PLOS ONE’s publication criteria as it currently stands. Therefore, we invite you to submit a revised version of the manuscript that addresses the points raised during the review process.

While the reviewer appreciate that most of the previous concerns were addressed, there are still several issues that need to be improved/addressed. Please revise your manuscript accordingly addressing all reviewer's comments.

We look forward to receiving your revised manuscript.

Kind regards,

Jan Rychtář

Academic Editor

PLOS ONE

Additional Editor Comments:

While the reviewer appreciates that most of the previous concerns were addressed, there are still several issues that need to be improved/addressed.

Reviewers' comments:

Reviewer's Responses to Questions

Comments to the Author

1. If the authors have adequately addressed your comments raised in a previous round of review and you feel that this manuscript is now acceptable for publication, you may indicate that here to bypass the “Comments to the Author” section, enter your conflict of interest statement in the “Confidential to Editor” section, and submit your "Accept" recommendation.

Reviewer #1: (No Response)

2. Is the manuscript technically sound, and do the data support the conclusions?

Reviewer #1: Partly

3. Has the statistical analysis been performed appropriately and rigorously?

Reviewer #1: Yes

4. Have the authors made all data underlying the findings in their manuscript fully available?

Reviewer #1: Yes

5. Is the manuscript presented in an intelligible fashion and written in standard English?

Reviewer #1: Yes

6. Review Comments to the Author

Reviewer #1: Thank you for addressing the majority of my questions and comments. However, I still have concerns about how the costs of false positives and false negatives are being estimated, which were further exposed while I was reviewing the changes made during this revision. I think additional clarification and/or justification is required for the term of the form (1-(1-θ)δ) in equations 12-15 in S2 File, which estimates the “probability of infection given random contact with an EVD patient”. The secondary attack rate θ is estimated in the relevant reference (Okware et al) as the fraction of contacts of an EVD case that get infected. Given that definition, I don’t understand why the duration of infectiousness (1/δ) is being used here in this way. I suppose if you’re assuming that the risk to each suspected case entering isolation is equivalent to the risk of coming into contact with a single EVD patient for one day, and you are assuming the published secondary attack rate is being estimated from household contacts who are exposed continuously for the entire duration of infectiousness, there is some possible justification for this equation, which would need to be further explained in the manuscript. Unfortunately I believe that this risk estimate cannot be appropriately justified using this reasoning- it’s not clear from the reference for the secondary attack rate value exactly how a contact is defined, and there are published estimates for the household attack rate that are much higher than 2.5% (see the meta-analysis in Dean et al. Clin Infect Dis 2016 https://doi.org/10.1093/cid/ciw114 which includes the reference cited here). A better way to estimate infection risk for false positives would probably be to use results from a study specifically designed to measure the risk of nosocomial infection in patients erroneously isolated in EVD units (e.g., Arkell et al Tropical Medicine and International Health 2016

https://doi.org/10.1111/tmi.12802) which seem to estimate a higher risk of infection than calculated in this manuscript (though the absolute risk of exposure is still reasonably low).

The estimate of risk of community transmission from a false negative (which was calculated using the same reasoning) is even more problematic. Again, the assumption being made seems to be that the expected number of cases resulting from a true EVD case being allowed to remain in the community is equivalent to the risk of a single person coming into contact with that EVD patient for one day, except now the secondary attack rate is 50% lower than was estimated previously (presumably to account for lower risk from “asymptomatic” EVD patients). This seems to me to be quite an underestimation of the expected number of cases. First, I think these patients are not truly “asymptomatic” and instead may have nonspecific symptoms or be presymptomatic, so I don’t think there is good justification for the assumption that their infectiousness is lower (especially by an arbitrary 50%, which the authors do not justify). Also, I’m guessing that many of these patients will return to their homes after a negative screening, potentially exposing multiple people (possibly in higher-risk caregiving roles) over multiple days, making the expected number of new cases resulting from a single false negative to be much higher than the 0.00125 calculated here. Using the R0 value for EVD community transmission (while perhaps a bit of an overestimate) for the effect of a false negative is likely more correct, and more easily justified, and would result in a false negative penalty that is orders of magnitude higher than the value currently used here.

Since these false negative/positive risks are very important factors in the cost-effectiveness analysis (and thus the conclusions of the whole manuscript), they require suitable justification before I can recommend publication. I apologize for not noticing these issues in my first review.

We appreciate your observation very much. We also appreciate the two informative documents you sent our way. We discovered further information in the articles that could offer trustworthy estimates of the expense and harm associated with false positives and false negatives.

According to the paper by Gilbert et al., this random probability (1-(1-θ)^δ) measures the risk of transmission during an infectious patient's entire period of time. We used this probability formula as such to compute the harm of errors in Ebola suspect-case classification. This probability is a function of both the secondary attack rate and the period or duration of infectiousness in an EVD patient. 

Once again, thank you; we believed that the SAR values presented in Dean et al.'s work should be the ones we must employ. As a result, we used SAR estimates from the study by Dean et al. in place of the SAR by Okware et al.

For the false-positive isolated case, we did not employ the SAR for contacts providing nursing care because we believed that a frontline vaccination for healthcare professionals that is implemented in Ebola outbreak control nowadays in health setting would be applied to protect them from professional transmission. Therefore, since only isolated suspects who are not immunized would be exposed to this danger trough physical contact, we applied the direct physical contact SAR. As a result, we used the mean household SAR, or 22.9% (11.6%-34.2%), for those with direct contact but no nursing.

Similarly, for the false-negative case, we assumed that no nursing care is done for mild or asymptomatic Ebola cases ruled out for the community and that the conclusion from Mbala et al. and Leroy et al. is that mild or non-specific symptomatic Ebola has a reduced role in person-to-person transmission. Therefore, we used the overall household SAR, which is 12.5% (8.6%–16.3%).

Using these estimates, we were able to re-estimate the arm related to errors of classification, e.g., false positive and false negative harm.

For the random probability formula's parameter, for instance, (1-(1-θ) ^δ), which is defined as 1/ the length of time that an EVD is infectious (1 divided by the length of time that an EVD is contagious). This detail is not clear in the supplemental appendix S2. Therefore, we should make this explicit in the supplemental appendix S2 file by clarifying each part in Eq. 12 to 15. 

Moreover, we did not include/use the R0 in our economic model. R0 is the number of secondary cases that one case would cause in a population that is completely susceptible, and it is estimated using biological, socio-behavioral, and environmental parameters that are involved in the transmission of the pathogen. As a result, R0 is calculated using ever-more complicated mathematical models, leading to incorrect interpretations and representations of R0 because each modeler develops models for specific objectives. Therefore, setting R0 numbers frequently given in the literature for historical epidemics may not be applicable to all Ebola outbreaks.

Additionally, R0 is one of the measures that is most frequently used to analyze the dynamics of infectious diseases, which was not the goal of this work.

We thus found it intuitive, simple to understand, and simple to generalize the random probability of contamination/transmission for false negatives (in the community) and false positives (in the healthcare setting) as a proportion of iatrogenic cases that could be generated in order to compute the errors of classification harm.

Therefore, we added these statements:

Method part of manuscript, on lines 221 to 224

As a frontline vaccination for healthcare workers would be implemented, we computed this probability of infection using the secondary attack rate (SAR) for direct physical contact of 22.9% (95% CI: 11.6%–34.2%) for those with direct contact but no nursing in the hospital (Ref. Dean et al.).

On lines 230 to 233

Therefore, we calculated the random probability that these false-negative cases generate Ebola cases in the community using the overall household SAR of 12.5% (95% CI: 8.6%–16.3 percentage) for human-to-human transmission in this setting without nursing (Ref. Dean et al.).

We corrected the harm associated with false positive and false negative in table 2 and in text.

In the results part of the manuscript:

We updated where needed accordingly (Tables 4 and 5, Figures 4 to 8, Supplementary Figures and the text of the manuscript

I have a couple additional minor comments:

- I think Eq. 13 in S2 File is missing a 1-Prev term?

Thank you very much. Yes, this term is missing in this formula. Thus, we corrected that as follows: 

Probability of iatrogenic EVD=[(1-〖Prev〗_EVD )×((1-Spe〖c_t〗_i )+Spe〖c_t〗_i×(1-Spe〖c_t〗_j ))×(1-(1 - θ_h)ᵟ)]

- Please explicitly state in the data availability statement that the raw data are available in the Supporting Outputs Data so readers know exactly where to find them.

Thank you very much.

Added on line 657 of data availability statement:

in supporting information-Compressed file (the Supporting Outputs Data).

- Figs. 7 and 8 legends: define red/green colors of points

Thank you very for this observation. 

We added in Fig. 7 and 8 legends and Suppl. Fig. legends these statements: 

Green points: ICERs that fall below the WTP line in Monte Carlo simulations, the maximum acceptable ICER (the algorithm is considered cost-effective); Red points: ICERs that fall above the WTP line, the maximum acceptable ICER (the algorithm is considered costly and less effective).

- It’s still a little unclear in the text what “joint” and “conditional” mean when referring to the ECPS test. The authors define these terms very clearly in their previous publication (ref. [26] in this manuscript, relevant passage copied below), and I’d suggest adding a similar description here or at least referencing this publication at the point in the text where these algorithms are defined.

“Finally, we evaluated our prediction models according to two additional clinical practice approaches in healthcare settings: joint and conditional tests or approaches. In both approaches, the suspects with no reported risk of exposure would be considered to not have the disease and the clinical team would act accordingly. No additional action, e.g., isolation, would be required. In the joint approach, the clinical team should clinically examine all suspects at low-, intermediate-, and high-risk reported exposure and recommend for isolation only those with a predicted probability of EVD greater than 5% (the cut-off chosen to maximize sensitivity, about 90 percent, in disease adverse context). In the conditional approach, the clinical team should isolate all suspects with high-risk reported exposure irrespective of their predicted probability of the disease and then suspects at low and intermediate reported exposure having an EVD-predicted probability greater than 5%.”

Thank you for this observation. This could make clear their use. Thank you.

We added these statements: 

Method part of the manuscript, on lines 155 to 164:

As described by Tshomba et al., the two screening methods—joint and conditional tests with ECPS—are methods in which suspects with no reported risk of exposure would be assumed to be free of the disease, and the clinical team would act appropriately (e.g., no further action is taken). In using the joint approach, all suspects at low-, intermediate-, and high-risk reported exposure are clinically assessed, and only those with a predicted likelihood of EVD greater than 5% are suggested for isolation. In the conditional test, regardless of their estimated probability of contracting the illness, all suspects with high-risk reported exposure should be isolated. Next, suspects with low and intermediate reported exposure who have an EVD-predicted probability of more than 5% should be isolated.________________________________________

7. PLOS authors have the option to publish the peer review history of their article (what does this mean?). If published, this will include your full peer review and any attached files.

Do you want your identity to be public for this peer review? For information about this choice, including consent withdrawal, please see our Privacy Policy.

Reviewer #1: Yes: Debra Van Egeren

Thank you. We analyzed them using PACE.

---

## [Decision Letter · Decision Letter 2]

31 Jul 2023

PONE-D-23-12291R2Cost-effectiveness of incorporating Ebola prediction score tools and rapid diagnostic tests into a screening algorithm: a decision analytic modelPLOS ONE

Dear Dr. Tshomba,

Thank you for submitting your manuscript to PLOS ONE. After careful consideration, we feel that it has merit but does not fully meet PLOS ONE’s publication criteria as it currently stands. Therefore, we invite you to submit a revised version of the manuscript that addresses the points raised during the review process.

The reviewer continues to raise a number of substantial issues. Given this is already a second revision, you will have only one more chance to revise the manuscript.

The reviewer is willing to communicate with you directly to go over the methodology rather than continue back and forth with the revisions.

If you are agreeable to this, please reach out to me directly via email (rychtarj@vcu.edu) and I will connect you with the reviewer.

We look forward to receiving your revised manuscript.

Kind regards,

Jan Rychtář

Academic Editor

PLOS ONE

Additional Editor Comments:

The reviewer continues to raise a number of substantial issues. Given this is already a second revision, you will have only one more chance to revise the manuscript.

The reviewer is willing to communicate with you directly to go over the methodology rather than continue back and forth with the revisions.

If you are agreeable to this, please reach out to me directly via email (rychtarj@vcu.edu) and I will connect you with the reviewer.

Reviewers' comments:

Reviewer's Responses to Questions

**Comments to the Author**

1. If the authors have adequately addressed your comments raised in a previous round of review and you feel that this manuscript is now acceptable for publication, you may indicate that here to bypass the “Comments to the Author” section, enter your conflict of interest statement in the “Confidential to Editor” section, and submit your "Accept" recommendation.

Reviewer #1: (No Response)

2. Is the manuscript technically sound, and do the data support the conclusions?

Reviewer #1: Partly

3. Has the statistical analysis been performed appropriately and rigorously? 

Reviewer #1: Yes

4. Have the authors made all data underlying the findings in their manuscript fully available?

Reviewer #1: Yes

5. Is the manuscript presented in an intelligible fashion and written in standard English?

Reviewer #1: Yes

6. Review Comments to the Author

Reviewer #1: Unfortunately, I think there are still two major issues with the core methodology of the study that have not been addressed.

First, I don’t think my concerns from the previous revision have been adequately addressed, so please allow me to clarify. I appreciate your response and your update to the SAR values used. However, I still believe that the formulae used to estimate the false positive and false negative costs do not accurately reflect the conceptual description of these costs given in the text or author response, do not correctly use the SAR values as consistently defined in both Gilbert et al and Dean et al, and may underestimate the expected number of cases resulting from a false positive/negative. The authors state that “[a]ccording to the paper by Gilbert et al., this random probability (1-(1-θ)^δ) measures the risk of transmission during an infectious patient's entire period of time” in their response. This is not consistent with the passage describing that expression in the Gilbert et al reference, which describes it as “the probability of infection given contact with an infectious individual” and uses it as a part of a differential equations SIR model. There are two differences between this definition and the authors’ interpretation. First, this expression represents risk of infection per contact, rather than the absolute risk for each infected individual regardless of contact rate. This is why, in this reference, this expression is multiplied by the contact rate. Second, the expression represents the probability of infection within one unit of time (here, 1 day or one contact), not the entire period of time (hence its inclusion in a system of differential equations). Therefore, by using this expression (without accounting for the number of contacts or amount of time each contact lasts) to represent the expected number of new infections that an infected individual will subsequently infect, the authors are implicitly assuming that each uninfected individual erroneously isolated in an EVD ward only experiences a single instance of direct contact with an infected patient (in the case of a FP) and infected individual erroneously screened as negative only has a single, transient contact after returning to the community (in the case of a FN). This assumption should be at least explicitly stated in the manuscript, and likely should be reevaluated, particularly for FNs. I can see how there would be little contact with infected individuals for uninfected patients in an EVD ward, but I have a hard time believing that an infected individual returning to a community after a false negative screening would have the same total risk over the entire period they are in the community as an infected individual with a single transient contact. That seems to be an underestimate, as I’m guessing most of these individuals will return home for multiple days and be in contact with multiple family members. However, the authors’ definition reproduced above describing the overall risk of transmission does not match with the expression they used but better matches either the SAR alone (defined in Gilbert et al as “the proportion of individuals who will become infected upon contact with an infectious individual during the total infectious period”, which agrees with the definition employed in Dean et al, the reference from which the parameter values were taken), which still doesn’t take into account the contact rate, or R0, which does take into account the contact rate and seems to be the closest metric to what the authors intend this cost to represent. As the authors correctly state in their response, “R0 is the number of secondary cases that one case would cause in a population that is completely susceptible”, which seems to exactly what is intended with this cost- the expected number of new cases resulting from a single infected individual returning to the community without isolation. I agree that there are issues with the estimation of R0 but the assumptions underlying the strategy the authors are currently using to estimate the FN risk is at the very least not justified in the manuscript and is likely underestimating the risk by multiple orders of magnitude. Admittedly it seems that with the current approach the FN risk hardly seems to affect the results at all (during the last revision this risk increased by about an order of magnitude and the results in Table 4 are almost the same), but I think this may reflect an issue with the overall approach as well, as outlined below.

Second, I have very serious concerns about how the effectiveness/payoff of a screening strategy is being defined more generally here. The authors repeatedly state that the effectiveness is defined as the number or fraction of true EVD cases isolated (e.g., in the abstract line 54, header row in Table 4, Methods lines 210-211); however, this is not the definition implemented in the payoff matrix (Table 3) or described later in the Methods (lines 213-215), where true positive and true negative outcomes are in fact weighted equally. The effectiveness is therefore nearly equal to the accuracy of the algorithm (i.e., (TP+TN)/(total screened)), with a negligible contribution from the penalties from FPs and FNs (which are 2 orders of magnitude smaller than the payoffs given to TN and TP, “by assumption”). This can lead to very problematic conclusions since the overwhelming majority of subjects being screened don’t have EVD (prevalence ~6% as given in Table 2), causing the specificity of the test to dominate the effectiveness metric. For example, consider the trivial screening algorithm where all individuals being screened are given a negative test result and sent home. This procedure has a sensitivity of 0 and specificity of 1, leading to an effectiveness payout of approximately 0.94 (calculated as (1-Prev)*(payoff of TN) + Prev*(payoff of FN)). This effectiveness value is higher than the best real algorithm tested in the manuscript! In fact, since the cost of each of these “screenings” would be very low (presumably $0), the methodology used in this manuscript would identify this dummy procedure as the best possible screening test in terms of cost, effectiveness, and the cost-effectiveness ratio. Obviously, this is undesirable behavior (just shutting down EVD isolation units is not a good strategy!). Therefore, it is necessary to come up with a different payoff matrix that doesn’t just mostly optimize the test specificity.

What would be a better payoff matrix? One option would be to implement the payoff matrix that is repeatedly described in the text but not actually used (“the number of isolated EVD cases (true positives)” lines 210-211), which would be assigning a value of 1 to TP and 0 to all other outcomes. Of course, this would just be proportional to the sensitivity of the screening algorithm and would not account for the specificity at all, so the strategy of isolating everyone who comes in would have the highest value (though also the highest cost). A better strategy, however, might be to quantify all payoffs in terms of monetary cost or benefit. This actually solves two serious issues with the current approach. First, it naturally solves the problem with deciding on how to weight TP and TN benefits that is explained above. TP events will have a benefit defined as the financial benefit to society for their isolation, which I’d suggest defining as the value generated by the effects of the patient receiving supportive care in the unit and extending their lifespan (e.g., see Bartsch et al Pathog Glob Health 2015 https://doi.org/10.1179%2F2047773214Y.0000000169 for estimates of the financial costs of EVD). TN events can be defined as having a payoff of $0. The second problem that this proposed redefinition into monetary units solves is the current mismatch in units between the TP or TN benefits and the FP and FN costs. TP and TN payoffs represent the number of correct screening calls made (assigning them each a value of 1) while FP and FN costs represent the number of additional expected EVD cases generated by the outcome (assigning a value of -1 per expected new EVD case). Therefore, the approach gives a correct negative screening result the same weight as generating a new EVD case that wouldn’t have occurred without an FP/FN result. These two events don’t seem like they should be on the same scale or be equivalent- causing a new infection that wouldn’t have otherwise occurred should be quite a bit more costly, I think! Instead, this current approach makes the FP and FN costs essentially negligible. Assigning FP and FN events monetary costs instead (perhaps by estimating the financial cost of each new EVD case) would put them on the same scale as TP/TN payoffs.

Overall, I think the conceptual issues and problematic limiting behavior of the current approach are serious and warrant major changes before publication. However, it’s possible that even after these changes, the conclusions of the manuscript will remain the same. Currently, the general conclusion seems to be that algorithms with higher specificity (i.e., those that use the ECPS as a joint or conditional test) perform better than lower specificity algorithms, not only because they have higher “effectiveness” as currently defined, but also because they have significantly lower cost since they isolate fewer people. The differences in specificity between these two groups of algorithms is so great (~35% vs. >80%) that the conclusion that these tests are more cost effective will almost certainly continue to hold.

7. PLOS authors have the option to publish the peer review history of their article (what does this mean?). If published, this will include your full peer review and any attached files.

Reviewer #1: **Yes: **Debra Van Egeren

---

## [Author Response · Author response to Decision Letter 2]

9 Sep 2023

Kinshasa, September 9, 2023

Tshomba Oloma Antoine

Institut National de Recherche Biomédicale (INRB) 

Kinshasa, Dem. Rep. of Congo, 

antotshomba@yahoo.fr

+243 815602451

September 9, 2023

Jan Rychtář

Academic Editor

PLOS ONE Journal

plosone@plos.org

Dear Editor,

We are resubmitting our manuscript entitled “Cost-effectiveness of incorporating Ebola prediction score tools and rapid diagnostic tests into a screening algorithm: a decision analytic model” as a Research Article for consideration of publication in the PLOS ONE Journal. 

First, we want to express our gratefulness to the Editor and Reviewers for their overall very positive comments on our work and their suggestions for improvement. Through this letter, we have tried answering to the best of our knowledge the questions, suggestions and remarks provided by the Editors and Reviewers. 

None of the authors has a competing interest to declare and our manuscript has not been submitted, or accepted elsewhere. All authors have contributed to, seen, and approved the final, submitted version of the manuscript.

We have upload the following documents:

- The clean version of the manuscript is in the file labeled “Manuscript.”

- The track changes version of the manuscript is in the file labeled "Revised Manuscript with Track Changes."

- The covering letter addressing the editorial and referees’ comments is in the file labeled "Response to Reviewers."

- We used secondary attack rates from the meta-analysis by Dean et al. and computed the FP payoff to include in our model, and we used the basic reproductive number to express the FN penalty.

- We made changes where needed in the manuscript text, figures, supporting materials, and data.

Again thank you a million for all deepest and helpful remarks and recommendations you have addressed on our manuscript. 

We will look forward to hearing whether this manuscript can be considered of interest for publication in the PLOS ONE Journal and remain at your disposal for any required clarifications. 

Yours sincerely,

Antoine Tshomba

PONE-D-23-12291R2

Cost-effectiveness of incorporating Ebola prediction score tools and rapid diagnostic tests into a screening algorithm: a decision analytic model

PLOS ONE

Dear Dr. Tshomba,

Thank you for submitting your manuscript to PLOS ONE. After careful consideration, we feel that it has merit but does not fully meet PLOS ONE’s publication criteria as it currently stands. Therefore, we invite you to submit a revised version of the manuscript that addresses the points raised during the review process.

The reviewer continues to raise a number of substantial issues. Given this is already a second revision, you will have only one more chance to revise the manuscript.

The reviewer is willing to communicate with you directly to go over the methodology rather than continue back and forth with the revisions.

If you are agreeable to this, please reach out to me directly via email (rychtarj@vcu.edu) and I will connect you with the reviewer.

We look forward to receiving your revised manuscript.

Kind regards,

Jan Rychtář

Academic Editor

PLOS ONE

Additional Editor Comments:

The reviewer continues to raise a number of substantial issues. Given this is already a second revision, you will have only one more chance to revise the manuscript.

The reviewer is willing to communicate with you directly to go over the methodology rather than continue back and forth with the revisions.

If you are agreeable to this, please reach out to me directly via email (rychtarj@vcu.edu) and I will connect you with the reviewer.

Reviewers' comments:

Reviewer's Responses to Questions

Comments to the Author

1. If the authors have adequately addressed your comments raised in a previous round of review and you feel that this manuscript is now acceptable for publication, you may indicate that here to bypass the “Comments to the Author” section, enter your conflict of interest statement in the “Confidential to Editor” section, and submit your "Accept" recommendation.

Reviewer #1: (No Response)

2. Is the manuscript technically sound, and do the data support the conclusions?

Reviewer #1: Partly

3. Has the statistical analysis been performed appropriately and rigorously?

Reviewer #1: Yes

4. Have the authors made all data underlying the findings in their manuscript fully available?

Reviewer #1: Yes

5. Is the manuscript presented in an intelligible fashion and written in standard English?

Reviewer #1: Yes

6. Review Comments to the Author

Reviewer #1: Unfortunately, I think there are still two major issues with the core methodology of the study that have not been addressed.

First, I don’t think my concerns from the previous revision have been adequately addressed, so please allow me to clarify. I appreciate your response and your update to the SAR values used. However, I still believe that the formulae used to estimate the false positive and false negative costs do not accurately reflect the conceptual description of these costs given in the text or author response, do not correctly use the SAR values as consistently defined in both Gilbert et al and Dean et al, and may underestimate the expected number of cases resulting from a false positive/negative. The authors state that “[a]ccording to the paper by Gilbert et al., this random probability (1-(1-θ)^δ) measures the risk of transmission during an infectious patient's entire period of time” in their response. This is not consistent with the passage describing that expression in the Gilbert et al reference, which describes it as “the probability of infection given contact with an infectious individual” and uses it as a part of a differential equations SIR model. There are two differences between this definition and the authors’ interpretation. First, this expression represents risk of infection per contact, rather than the absolute risk for each infected individual regardless of contact rate. This is why, in this reference, this expression is multiplied by the contact rate. Second, the expression represents the probability of infection within one unit of time (here, 1 day or one contact), not the entire period of time (hence its inclusion in a system of differential equations). Therefore, by using this expression (without accounting for the number of contacts or amount of time each contact lasts) to represent the expected number of new infections that an infected individual will subsequently infect, the authors are implicitly assuming that each uninfected individual erroneously isolated in an EVD ward only experiences a single instance of direct contact with an infected patient (in the case of a FP) and infected individual erroneously screened as negative only has a single, transient contact after returning to the community (in the case of a FN). This assumption should be at least explicitly stated in the manuscript, and likely should be reevaluated, particularly for FNs. I can see how there would be little contact with infected individuals for uninfected patients in an EVD ward, but I have a hard time believing that an infected individual returning to a community after a false negative screening would have the same total risk over the entire period they are in the community as an infected individual with a single transient contact. That seems to be an underestimate, as I’m guessing most of these individuals will return home for multiple days and be in contact with multiple family members. However, the authors’ definition reproduced above describing the overall risk of transmission does not match with the expression they used but better matches either the SAR alone (defined in Gilbert et al as “the proportion of individuals who will become infected upon contact with an infectious individual during the total infectious period”, which agrees with the definition employed in Dean et al, the reference from which the parameter values were taken), which still doesn’t take into account the contact rate, or R0, which does take into account the contact rate and seems to be the closest metric to what the authors intend this cost to represent. As the authors correctly state in their response, “R0 is the number of secondary cases that one case would cause in a population that is completely susceptible”, which seems to exactly what is intended with this cost- the expected number of new cases resulting from a single infected individual returning to the community without isolation. I agree that there are issues with the estimation of R0 but the assumptions underlying the strategy the authors are currently using to estimate the FN risk is at the very least not justified in the manuscript and is likely underestimating the risk by multiple orders of magnitude. Admittedly it seems that with the current approach the FN risk hardly seems to affect the results at all (during the last revision this risk increased by about an order of magnitude and the results in Table 4 are almost the same), but I think this may reflect an issue with the overall approach as well, as outlined below.

Thank you very much, and I really appreciate this comment. We feel that not taking into account the average number of contacts that these false negatives mistakenly return in the community will not give the correct reflection of the number of cases that these false negatives could generate in the community. 

We think our model that favors the specificity over the sensitivity of screening tests employed in tested algorithms results from underestimating the harm associated with FN.

In conclusion, you are right about this fundamental observation, and we sincerely thank you for it.

Therefore, to correct the problem with our model, we are going to use the estimated value of the penalty for FP, as it appears easy to estimate both the transmissibility and the average number of healthy people exposed in isolation due to this misclassification. With the information available, we will multiply the strength of infection and estimating the average number of possible contacts (here, the number of people exposed) per isolation of a false positive to score this error of classification (FP). 

However, estimating the number of contacts that this false negative could have in the community remains difficult to pin down. Thus, the use of Ro, a composite measure that gives both information on transmissibility and average contact numbers, may represent the way to solve the problem linked to the underestimation of the FN penalty, thus balancing the favor given to the specificity of our present model.

In short, to correct the problem with our model, we are going to use the estimated value of the penalty for FP, as it appears easy to estimate both the transmissibility and the average number of healthy people exposed in isolation due to this misclassification. However, as far as FN is concerned, we intend to use the Ro value in the absence of any intervention that can be found in the literature. 

 Thus, in the method part of the manuscript, in the paragraph regarding the classification error harming computation, we corrected and rewrote sentences 

• On lines from 127 to 128, as follows: 

We negatively assigned this probability reported to the number of non-EVD exposed contacts due to this classification error in the isolation (as a payoff). Negatively because it is the harm caused by isolation, e.g., iatrogenic harm. For this erroneously false positive isolated, we assumed that each isolated false positive and his two family caregivers were non-EVD (e.g., three non-EVD would be exposed in the isolation ward). 

• On lines 236 to 238, as follows:

Therefore, we assigned a score equal to minus the anticipated number of Ebola cases that this false-negative case—which was ruled out—would produce in the entire susceptible population (e.g., minus the basic reproductive number, the Ro, which accounts for the transmissibility and the typical number of community contacts that this false-negative would harm)

• We added these statements on lines 239 to 246:

For this erroneously false positive isolated, we assumed that each isolated false positive and his two family caregivers were non-EVD (thus, three non-EVD in the isolation ward). Thus, a value of -0.077 was assigned to each isolated non-EVD case. We hypothesized that the community as a whole would be exposed to the Ebola virus infection by these false negatives in the community. Therefore, we assigned a score equal to minus the anticipated number of Ebola cases that this false-negative case—which was ruled out—would produce in the entire susceptible population (e.g., minus the basic reproductive number, the Ro, which accounts for the transmissibility and the typical number of community contacts that this false-negative would harm and represents ). In a population that is entirely susceptible, the basic reproduction number is the number of secondary instances that one case would result in.

For each EVD case ruled out, we assigned a value of -2.49, e.g., minus the Ro as estimated by Lewnard [http://dx.doi.org/10.1016/S1473-3099(14)70995-8], as the effectiveness payoff.

Second, I have very serious concerns about how the effectiveness/payoff of a screening strategy is being defined more generally here. The authors repeatedly state that the effectiveness is defined as the number or fraction of true EVD cases isolated (e.g., in the abstract line 54, header row in Table 4, Methods lines 210-211); however, this is not the definition implemented in the payoff matrix (Table 3) or described later in the Methods (lines 213-215), where true positive and true negative outcomes are in fact weighted equally. The effectiveness is therefore nearly equal to the accuracy of the algorithm (i.e., (TP+TN)/(total screened)), with a negligible contribution from the penalties from FPs and FNs (which are 2 orders of magnitude smaller than the payoffs given to TN and TP, “by assumption”). This can lead to very problematic conclusions since the overwhelming majority of subjects being screened don’t have EVD (prevalence ~6% as given in Table 2), causing the specificity of the test to dominate the effectiveness metric. For example, consider the trivial screening algorithm where all individuals being screened are given a negative test result and sent home. This procedure has a sensitivity of 0 and specificity of 1, leading to an effectiveness payout of approximately 0.94 (calculated as (1-Prev)*(payoff of TN) + Prev*(payoff of FN)). This effectiveness value is higher than the best real algorithm tested in the manuscript! In fact, since the cost of each of these “screenings” would be very low (presumably $0), the methodology used in this manuscript would identify this dummy procedure as the best possible screening test in terms of cost, effectiveness, and the cost-effectiveness ratio. Obviously, this is undesirable behavior (just shutting down EVD isolation units is not a good strategy!). Therefore, it is necessary to come up with a different payoff matrix that doesn’t just mostly optimize the test specificity.

What would be a better payoff matrix? One option would be to implement the payoff matrix that is repeatedly described in the text but not actually used (“the number of isolated EVD cases (true positives)” lines 210-211), which would be assigning a value of 1 to TP and 0 to all other outcomes. Of course, this would just be proportional to the sensitivity of the screening algorithm and would not account for the specificity at all, so the strategy of isolating everyone who comes in would have the highest value (though also the highest cost). A better strategy, however, might be to quantify all payoffs in terms of monetary cost or benefit. This actually solves two serious issues with the current approach. First, it naturally solves the problem with deciding on how to weight TP and TN benefits that is explained above. TP events will have a benefit defined as the financial benefit to society for their isolation, which I’d suggest defining as the value generated by the effects of the patient receiving supportive care in the unit and extending their lifespan (e.g., see Bartsch et al Pathog Glob Health 2015 https://doi.org/10.1179%2F2047773214Y.0000000169 for estimates of the financial costs of EVD). TN events can be defined as having a payoff of $0. The second problem that this proposed redefinition into monetary units solves is the current mismatch in units between the TP or TN benefits and the FP and FN costs. TP and TN payoffs represent the number of correct screening calls made (assigning them each a value of 1) while FP and FN costs represent the number of additional expected EVD cases generated by the outcome (assigning a value of -1 per expected new EVD case). Therefore, the approach gives a correct negative screening result the same weight as generating a new EVD case that wouldn’t have occurred without an FP/FN result. These two events don’t seem like they should be on the same scale or be equivalent- causing a new infection that wouldn’t have otherwise occurred should be quite a bit more costly, I think! Instead, this current approach makes the FP and FN costs essentially negligible. Assigning FP and FN events monetary costs instead (perhaps by estimating the financial cost of each new EVD case) would put them on the same scale as TP/TN payoffs.

Overall, I think the conceptual issues and problematic limiting behavior of the current approach are serious and warrant major changes before publication. However, it’s possible that even after these changes, the conclusions of the manuscript will remain the same. Currently, the general conclusion seems to be that algorithms with higher specificity (i.e., those that use the ECPS as a joint or conditional test) perform better than lower specificity algorithms, not only because they have higher “effectiveness” as currently defined, but also because they have significantly lower cost since they isolate fewer people. The differences in specificity between these two groups of algorithms is so great (~35% vs. >80%) that the conclusion that these tests are more cost effective will almost certainly continue to hold.

Thank you very much for this comment.

As we just pointed out above, we recognized that our model underestimated the penalties, especially for the FN, leading to a model that only favors the specificity of our tested screening algorithms. We corrected this situation by taking into account the average number of non-EVD cases that would be exposed to the disease because of these classification errors (in the isolation ward or community setting).

In addition, using the monetary valuation of effectiveness means that we should use cost-benefit evaluation.

Yes, we know! Cost-benefit analysis (CBA) is the reference of economic studies as it is the most comprehensive and theoretically sound form of economic evaluation, and it has been used as an aid to decision-making in many different areas of economic and social policy in the public sector. But, in this manuscript, our objective was rather to evaluate the cost-effectiveness of the screening algorithms we tested; thus, we did not use this cost-benefit analysis (CBA) method, which estimates and totals up the equivalent monetary value of the benefits and costs relative to each screening algorithm. Thus, it seeks to place monetary values on both the inputs (costs) and outcomes (benefits) of health care to establish whether they are worthwhile. CBA requires program consequences to be valued in monetary units, thus enabling the analyst to make a direct comparison of the program's incremental cost with its incremental consequences in commensurate units of measurement, e.g., dollars or pounds. 

We chose CEA because, in this study, we only evaluated one pillar of the Ebola control strategies: the screening of Ebola suspects. We did not analyze all the other pillars of response to the Ebola epidemic, e.g., Ebola-specific treatments for positive patients. 

Again, not much is known about the Ebola disease yet. Information about the monetary value of each outcome is not yet available. We do not think that performing a CBA will be easy or plausible, as no extensive analysis of their monetary value exists currently. 

In short, we are aware of the problem with CEA and CUA, as the numerator and numerator do not have the same unit, and performing a CBA has the problem of a lack of monetary data to date. 

Therefore, we chose the CEA that, easily, can be performed with the couple of data sets available. 

When more information on the monetary value of effectiveness becomes available, a CBA will be worthwhile in economic evaluation and this will be another manuscript to write. 

At least, we tried to perform by including inputs according to the other scenarios you proposed (e.g., assigning a value of 1 to TP and 0 to all other outcomes or assigning -1 per expected new EVD case), but this resulted in implausible results of effectiveness for all algorithms (mostly quite a zero or negative number as effectiveness).

To sum:

Sincerely, we are grateful for all the comments you have made on this manuscript. All your comments were fundamental and really improved our manuscript's quality.________________________________________

7. PLOS authors have the option to publish the peer review history of their article (what does this mean?). If published, this will include your full peer review and any attached files.

Do you want your identity to be public for this peer review? For information about this choice, including consent withdrawal, please see our Privacy Policy.

Reviewer #1: Yes: Debra Van Egeren________________________________________

Thank you.

We uploaded and analyzed them using PACE before the submission

---

## [Decision Letter · Decision Letter 3]

27 Sep 2023

PONE-D-23-12291R3Cost-effectiveness of incorporating Ebola prediction score tools and rapid diagnostic tests into a screening algorithm: a decision analytic modelPLOS ONE

Dear Dr. Tshomba,

Thank you for submitting your manuscript to PLOS ONE. After careful consideration, we feel that it has merit but does not fully meet PLOS ONE’s publication criteria as it currently stands. Therefore, we invite you to submit a revised version of the manuscript that addresses the points raised during the review process.

The reviewer is happy with the revision and recommends only minor changes before the manuscript can be accepted for the publication.

We look forward to receiving your revised manuscript.

Kind regards,

Jan Rychtář

Academic Editor

PLOS ONE

Journal Requirements:

Additional Editor Comments:

The reviewer is happy with the revision and recommends only minor changes before the manuscript can be accepted for the publication.

Reviewers' comments:

Reviewer's Responses to Questions

**Comments to the Author**

1. If the authors have adequately addressed your comments raised in a previous round of review and you feel that this manuscript is now acceptable for publication, you may indicate that here to bypass the “Comments to the Author” section, enter your conflict of interest statement in the “Confidential to Editor” section, and submit your "Accept" recommendation.

Reviewer #1: All comments have been addressed

2. Is the manuscript technically sound, and do the data support the conclusions?

Reviewer #1: Yes

3. Has the statistical analysis been performed appropriately and rigorously? 

Reviewer #1: Yes

4. Have the authors made all data underlying the findings in their manuscript fully available?

Reviewer #1: Yes

5. Is the manuscript presented in an intelligible fashion and written in standard English?

Reviewer #1: Yes

6. Review Comments to the Author

Reviewer #1: At this point I think the manuscript can be published if the language describing the definition of effectiveness is corrected, and I do not think I need to see this manuscript again. In the abstract (and in many other places in the text, all of which need to be changed), the effectiveness is described as follows:

Our analysis found dual ECPS as a conditional test with the QuickNavi™-Ebola RDT algorithm to be the most cost-effective screening algorithm for EVD, with an effectiveness of 0.86 (e.g., isolating 86% of EVD cases).

The last part ("isolating 86% of EVD cases") is incorrect. The provided definition (isolating X% of true cases) I quoted above is actually the sensitivity of the test, not the effectiveness. The effectiveness formula now being used still has no real-world meaning, but I believe it is mathematically equivalent to using the expected number of EVD cases prevented per individual screened (do NOT use this as the definition either though, as stated this is not the actual definition of that number as it stands). I would instead not provide any real-world definition and simply indicate that it is a metric of test effectiveness.

7. PLOS authors have the option to publish the peer review history of their article (what does this mean?). If published, this will include your full peer review and any attached files.

Reviewer #1: **Yes: **Debra Van Egeren

---

## [Author Response · Author response to Decision Letter 3]

4 Oct 2023

Kinshasa, October 3, 2023

Tshomba Oloma Antoine

Institut National de Recherche Biomédicale (INRB) 

Kinshasa, Dem. Rep. of Congo, 

antotshomba@yahoo.fr

+243 815602451

October 3, 2023

Jan Rychtář

Academic Editor

PLOS ONE Journal

plosone@plos.org

Dear Editor,

We are resubmitting our manuscript entitled “Cost-effectiveness of incorporating Ebola prediction score tools and rapid diagnostic tests into a screening algorithm: a decision analytic model” as a Research Article for consideration of publication in the PLOS ONE Journal. 

First, we want to express our gratefulness to the Editor and Reviewers for their overall very positive comments on our work and their suggestions for improvement. Through this letter, we have tried answering to the best of our knowledge the questions, suggestions and remarks provided by the Editors and Reviewers. 

None of the authors has a competing interest to declare and our manuscript has not been submitted, or accepted elsewhere. All authors have contributed to, seen, and approved the final, submitted version of the manuscript.

We have upload the following documents:

- The clean version of the Manuscript, file labeled "Manuscript"

- The track changes version of the manuscript, file labeled “Revised Manuscript with Track Changes”. 

- The covering letter addressing the editorial and referees’ comments, file labeled “Response to Reviewers”.

- We corrected the effectiveness’s definition and made changes where needed in the manuscript text, and figures’ labels 

Frankly, we would like thank all of you for your fundamental and helpful remarks and recommendations you have addressed on our manuscript. 

We will look forward to hearing whether this manuscript can be considered of interest for publication in the PLOS ONE Journal and remain at your disposal for any required clarifications. 

Yours sincerely,

Antoine Tshomba

From: PLOS ONE <em@editorialmanager.com>

À :Antoine Oloma Tshomba

mer. 27 sept. à 11:46

PONE-D-23-12291R3

Cost-effectiveness of incorporating Ebola prediction score tools and rapid diagnostic tests into a screening algorithm: a decision analytic model

PLOS ONE

Dear Dr. Tshomba,

Thank you for submitting your manuscript to PLOS ONE. After careful consideration, we feel that it has merit but does not fully meet PLOS ONE’s publication criteria as it currently stands. Therefore, we invite you to submit a revised version of the manuscript that addresses the points raised during the review process.

The reviewer is happy with the revision and recommends only minor changes before the manuscript can be accepted for the publication.

We look forward to receiving your revised manuscript.

Kind regards,

Jan Rychtář

Academic Editor

PLOS ONE

Journal Requirements:

Thank you very much. We reviewed all our references for retracted papers, but we did not find any. 

Additional Editor Comments:

The reviewer is happy with the revision and recommends only minor changes before the manuscript can be accepted for the publication.

Reviewers' comments:

Reviewer's Responses to Questions

Comments to the Author

1. If the authors have adequately addressed your comments raised in a previous round of review and you feel that this manuscript is now acceptable for publication, you may indicate that here to bypass the “Comments to the Author” section, enter your conflict of interest statement in the “Confidential to Editor” section, and submit your "Accept" recommendation.

Reviewer #1: All comments have been addressed

2. Is the manuscript technically sound, and do the data support the conclusions?

Reviewer #1: Yes

3. Has the statistical analysis been performed appropriately and rigorously?

Reviewer #1: Yes

4. Have the authors made all data underlying the findings in their manuscript fully available?

Reviewer #1: Yes

5. Is the manuscript presented in an intelligible fashion and written in standard English?

Reviewer #1: Yes

6. Review Comments to the Author

Reviewer #1: At this point I think the manuscript can be published if the language describing the definition of effectiveness is corrected, and I do not think I need to see this manuscript again. In the abstract (and in many other places in the text, all of which need to be changed), the effectiveness is described as follows:

Our analysis found dual ECPS as a conditional test with the QuickNavi™-Ebola RDT algorithm to be the most cost-effective screening algorithm for EVD, with an effectiveness of 0.86 (e.g., isolating 86% of EVD cases).

The last part ("isolating 86% of EVD cases") is incorrect. The provided definition (isolating X% of true cases) I quoted above is actually the sensitivity of the test, not the effectiveness. The effectiveness formula now being used still has no real-world meaning, but I believe it is mathematically equivalent to using the expected number of EVD cases prevented per individual screened (do NOT use this as the definition either though, as stated this is not the actual definition of that number as it stands). I would instead not provide any real-world definition and simply indicate that it is a metric of test effectiveness.

Thank you very much for all these fundamental observations. 

We agree with you! Actually, mathematically, the effectiveness relative to the design does not have a real-world meaning. 

Indeed, as we assigned and weighted equally true positive and true-negative outcomes, the fraction of effectiveness can represent the number of suspects correctly classified in the total number of suspects screened (TP and TN in the total number of suspects screened). However, as we considered the consequences of the corresponding misclassification errors (FP and FN) that we assigned negatively in the formulas as harm, the effectiveness fraction accurately approximates the accuracy, but not exactly. 

In short, we can say that the effectiveness number reported in the manuscript can be interpreted as the “net number of patients correctly classified.” (i.e., the accuracy minus the penalty)

So, we added these statements to define the effectiveness on lines 375 to 378 as follows: 

This fraction of effectiveness reflects the number of EVD suspects who were correctly classified after taking into consideration the harm brought on by incorrect classifications. It can be seen as the percentage of patients who were correctly categorized for each patient screened.

Additionally, we made changes where needed in the manuscript text, and figures’ labels 

7. PLOS authors have the option to publish the peer review history of their article (what does this mean?). If published, this will include your full peer review and any attached files.

Do you want your identity to be public for this peer review? For information about this choice, including consent withdrawal, please see our Privacy Policy.

Reviewer #1: Yes: Debra Van Egeren

We uploaded and checked them on PACE

---

## [Editor Report · Decision Letter 4]

5 Oct 2023

Cost-effectiveness of incorporating Ebola prediction score tools and rapid diagnostic tests into a screening algorithm: a decision analytic model

PONE-D-23-12291R4

Dear Dr. Tshomba,

We’re pleased to inform you that your manuscript has been judged scientifically suitable for publication and will be formally accepted for publication once it meets all outstanding technical requirements.

Kind regards,

Jan Rychtář

Academic Editor

PLOS ONE
---

## [Editor Report · Acceptance letter]

9 Oct 2023

PONE-D-23-12291R4 

Cost-effectiveness of incorporating Ebola prediction score tools and rapid diagnostic tests into a screening algorithm: a decision analytic model 

Dear Dr. Tshomba:

I'm pleased to inform you that your manuscript has been deemed suitable for publication in PLOS ONE. Congratulations! Your manuscript is now with our production department. 

Kind regards, 

on behalf of

Dr. Jan Rychtář 

Academic Editor

PLOS ONE